# LINA: Exploring Linear Autoregressive Image Generation with Continuous Tokens

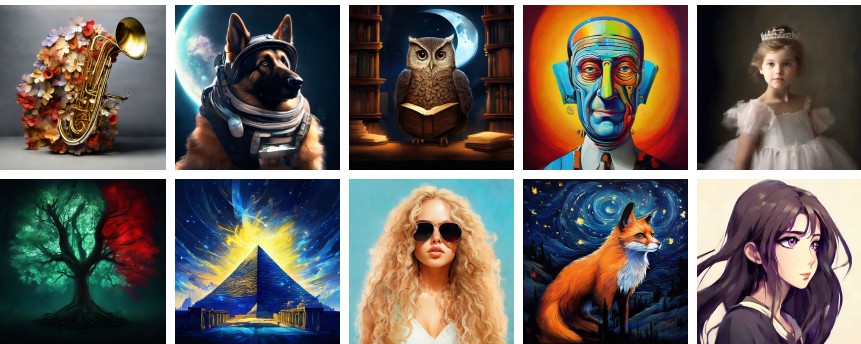

Figure 1: Qualitative results of 1024px samples powered by LINA.

## Abstract

In this paper, we systematically explore how efficient linear attention mechanisms, as alternatives to full attention, should be designed in the context of autoregressive image generative models with continuous tokens. We offer **LINA**, a simple and empirically validated text-to-image generation baseline built on pure linear attention, capable of rapidly generating high fidelity 1024×1024 images from user instructions. Our key contributions are: (1) An empirical study on scaling behavior w.r.t. parameter counts. We examine two crucial design choices: (i) normalization paradigms—division-based *vs.*subtraction-based, and (ii) the use of a depthwise convolution module on image features for locality capacity enhancement. We then compare which variants of linear attention scale most effectively in autoregressive image generation. (2) KV gate mechanism: We empirically find that applying data-independent learnable parameters to the key and value states, thereby enabling flexible memory management and benefiting sample quality. We run thorough ablations on the KV gate design choices and visually show off the patterns it learns. Building on these designs, LINA achieves promising results on both class-conditional (C2I) and text-to-image (T2I) generation. Compared to diffusion models of similar scale and autoregressive models with full attention, LINA delivers competitive performance: FID of 2.18 on ImageNet (∼1.4B) and an overall score of 0.74 on GenEval (∼1.5B). Code and models will be open-sourced.

## 1 Introduction

The field of image generation is evolving rapidly (Ho et al., 2020; Peebles & Xie, 2023; Chen et al., 2024b;a). Recently, a method based on autoregressive models with continuous tokens (Li et al., 2024b) has shown promising potential. As an emerging paradigm, it has not only been validated in text-to-image tasks (Fan et al., 2025a), but has also inspired video generation (Deng et al., 2025) and been scaled up to large models (*e.g.*, 14B parameters (Team et al., 2025)). Despite its appeal, one major bottleneck is efficiency: the approach involves both multi-step autoregression and diffusion, making it less practical for long-sequence generation such as high-resolution images or long videos.

A natural direction is to focus on the model side, turning to efficient linear attention mechanism (Katharopoulos et al., 2020), which has already been extensively explored in both vision transformer (ViT) (Dosovitskiy et al., 2020) based perception models (Cai et al., 2023; Han et al., 2023)

and diffusion transformer (DiT) (Peebles & Xie, 2023) based generative models (Xie et al., 2025a;b; Wang et al., 2025). Unlike ViTs and DiTs, autoregressive models carry out inference in a different manner: image tokens are generated step by step, with the set of known tokens gradually expanding as inference progresses. This distinction suggest that the design philosophy for linear attention may need a careful reconsideration, a topic that often remains undisclosed in the literature. Accordingly, we focus on the question: *What design of linear attention suits autoregressive image generative models?* We make two main contributions to facilitate future study:

**An empirical study on parameter scaling of linear attention in image generation.** We argue that for linear attention, *normalization paradigm* and *locality augmentation* are two key factors that affects the scaling behavior w.r.t. *parameter counts*. On one hand, normalization serves as a fundamental component of deep neural networks; in the attention mechanism, it numerically ensures that the attention scores sum to one. On the other hand, linear attention, compared with full attention, is well known for its insufficiency in locality (Han et al., 2023; 2024). We apply depthwise convolution (DWC) (Chollet, 2017; Howard et al., 2017) directly to image features only, as a compensation for the insufficient locality. In Sec. 4.2, we conduct a systematic empirical study to explore the following design choices.

- *Design choices 1: Division-based or subtraction-based normalization for linear attention.*
- *Design choices 2: Whether image tokens should be locality augmented in linear attention.*

The above design choices lead to four configurations. For each, we train models with three different parameter counts: ∼0.4B, ∼0.6B, and ∼1.4B. The experiments are conducted on the ImageNet (Deng et al., 2009) $256 \times 256$ class-conditional image generation task. We use FID (Heusel et al., 2017), sFID (Nash et al., 2021), Inception Score (Salimans et al., 2016), and Precision/Recall (Kynkäänniemi et al., 2019) as evaluation metrics, and report how sampling quality scales with parameter counts. The results were surprising to us: the injectivity brought by subtraction-based normalization turned out to be *not* crucial for autoregressive image generation, although injectivity has been underscored in perception models (Han et al., 2024). Moreover, the inductive bias introduced by DWC does *not* show limitations on scaling w.r.t. parameter counts.

**KV gate for memory management in linear attention for image generation.** In Sec. 4.3, we empirically explore gating mechanism (Yang et al., 2024; 2025; Zhang et al., 2024; Lin et al., 2025; Qiu et al., 2025) within bidirectional linear attention on autoregressive image generative models. We introduce *KV gate*, a simple yet effective mechanism for flexible memory management. The KV gate are *data-independent*, scalar-valued *learnable parameters* with *no* explicit range constraints. Ablation studies on the KV gate reveal that proper management of *both the memory and the normalization term* is crucial. We further provide detailed visualizations of the KV gate. As a result, our KV gate can serve as a readily applicable practice for linear image generative transformers.

Based on these explorations, our final model, LINA, is a linear autoregressive image generator that employs *division-based normalization* linear attention enhanced with a *DWC module* for locality enhancement, together with the *KV gate*. LINA is validated on class-conditional image generation (achieving an FID of 2.18 on the ImageNet $256 \times 256$ benchmark) and also adapts to text-to-image generation, where it can follow user instructions to efficiently produce high-fidelity images up to 1024px. LINA achieves a GenEval overall score of 74. Linear attention in LINA reduces FLOPs by ∼61% compared with full attention, showing the efficiency of our approach.

## 2 RELATED WORK

We briefly review related work in our main paper. Our study builds on autoregressive models with continuous tokens (Li et al., 2024b; Fan et al., 2025a), which have become a mainstream approach in image generation in recent years. This line of research has been explored not only with 14B-parameter autoregressive models (*e.g.*, NextStep-1 (Team et al., 2025)), but also successfully extended to video generation (*e.g.*, NOVA (Deng et al., 2025)). One of the key bottlenecks of such methods lies in their computational efficiency. To this end, we draw inspiration from efficient linear attention (Katharopoulos et al., 2020; Choromanski et al., 2021), which has already been successfully applied to both LLMs (Yang et al., 2024) and visual perception task (Cai et al., 2023; Han et al.,

2023). Along this line, several studies (Xie et al., 2025a;b; Wang et al., 2025; Zhu et al., 2024; Pu et al., 2024) have studied designing eficient DiTs with linear attention. Differently, we thoroughly discuss how linear attention should be designed in autoregressive image generative models.

## 3 PRELIMINARY

### 3.1 AUTOREGRESSIVE IMAGE GENERATION WITH CONTINUOUS TOKENS

Given a target of $N$ tokens $\{X_1, \dots, X_N\}$ to predict, masked autoregressive models (MAR) (Li et al., 2024b) complete the prediction task in $K$ steps. Illustrated in Fig. 2-(a), at every step $k$, a random-order autoregressive model predicts a set of tokens $\mathbf{S}_k = \{X_i, X_{i+1}, \dots, X_j\}$ with $\cup_k \mathbf{S}_k = \{X_1, \dots, X_N\}$, conditioned on the previously generated tokens $\{X_1, \dots, X_{i-1}\}$:

$$p(X_1, \dots, X_N) = p(\mathbf{S}_1, \dots, \mathbf{S}_K) = \prod_{k}^{K} p(\mathbf{S}_k \mid \mathbf{S}_1, \dots, \mathbf{S}_{k-1}). \tag{1}$$

Typically, MAR models consist of a network (*e.g.*, Transformer (Vaswani et al., 2017)) that predicts a condition vector from the input, and a diffusion (Ho et al., 2020; Sohl-Dickstein et al., 2015) or flow matching (Lipman et al., 2023; Liu et al., 2023) head (*e.g.*, MLP) that models the next token distribution conditioned on this vector.

Inference efficiency is a key research direction for MAR models, as both the autoregressive and diffusion processes require multiple iterations. In our work, we delve deep into the architecture design and explore how to leverage efficient linear attention in this context.

### 3.2 LINEAR ATTENTION

Formally, given an input sequence $I \in \mathbb{R}^{N \times D}$ of length $N$, we denote the queries, keys, and values in linear attention (Katharopoulos et al., 2020) by $Q, K, V \in \mathbb{R}^{N \times D}$. We refer to the kernel function as $\phi(\cdot)$ (*e.g.*, ReLU) and the output as $O \in \mathbb{R}^{N \times D}$ (for simplicity, we assume attention head as 1). Linear attention introduces a normalization factor $\gamma \in \mathbb{R}^N$ that ensures the normalized attention weights for each token $i$ sum to 1. Based on how this normalization term is defined, we categorize linear attention as follows[1].

**Division-based normalization.** In this formulation, the normalization factor in linear attention is placed in the denominator, akin to the softmax operation in full attention (Vaswani et al., 2017):

$$O_i^{(\mathrm{d})} = \sum_{j=1}^{N} \frac{A_{ij}^{(\mathrm{d})}}{\gamma_i^{(\mathrm{d})}} V_j = \sum_{j=1}^{N} \frac{\phi(Q_i)\phi(K_j)^{\top}}{\sum_{m=1}^{N} \phi(Q_i)\phi(K_m)^{\top}} V_j = \frac{\phi(Q_i)\left(\sum_{j=1}^{N} \phi(K_j)^{\top} V_j\right)}{\phi(Q_i)\left(\sum_{m=1}^{N} \phi(K_m)^{\top}\right)}, \tag{2}$$

where $\gamma_i^{(\mathrm{d})} \in \mathbb{R}$ denotes the scalar-valued division-based normalization term, calculated from $Q_i \in \mathbb{R}^{1 \times D}$ and the set of key states $K_m \in \mathbb{R}^{1 \times D}$ for $m \in [1, N]$. Linear attention reduces the computational complexity from $\mathcal{O}(N^2)$ in full attention to $\mathcal{O}(N)$, since for every query $Q_i$, both the memory $M = \sum_{j=1}^{N} \phi(K_j)^{\top} V_j \in \mathbb{R}^{D \times D}$ and $z = \sum_{m=1}^{N} \phi(K_m)^{\top} \in \mathbb{R}^{D \times 1}$ are shared and thus need to be computed only once.

**Subtraction-based normalization.** In this form, the normalization term is introduced as a distinct term, imparting linear attention with an injective property (Han et al., 2024):

---

[1]Here our terminology is based on (Han et al., 2024) and (Fan et al., 2025b).

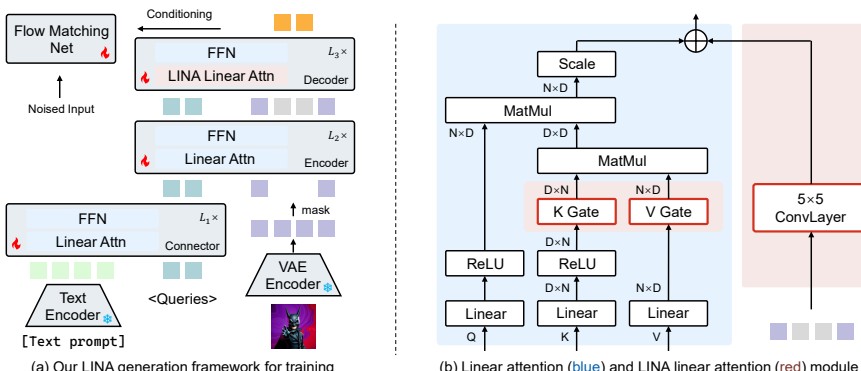

(a) Our LINA generation framework for training

(b) Linear attention (blue) and LINA linear attention (red) module

Figure 2: **Overview of LINA**: Fig. (a) illustrates the training pipeline, with a *Connector* for extracting text information, an *Encoder* to extract unmasked tokens, and a *Decoder* to reconstruct masked tokens for conditioning. A denoising flow matching MLP is used to sample tokens. Fig. (b) shows the *division-based normalization* linear attention, and our introduced *DWC* and *KV gate* (Sec. 4).

$$O_i^{(\mathrm{s})} = \sum_{j=1}^N \left( A_{ij}^{(\mathrm{s})} - \gamma_i^{(\mathrm{s})} \right) V_j = \sum_{j=1}^N \left[ \phi(Q_i) \frac{1}{N} \phi(K_j)^\top - \left( \frac{1}{N} \sum_{m=1}^N \phi(Q_i) \frac{1}{N} \phi(K_m)^\top - \frac{1}{N} \right) \right] V_j$$

$$= \phi(Q_i) \left( \frac{1}{N} \sum_{j=1}^N \phi(K_j)^\top V_j \right) - \left( \phi(Q_i) \frac{1}{N} \sum_{m=1}^N \phi(K_m)^\top - 1 \right) \frac{1}{N} \sum_{j=1}^N V_j,$$

$$(3)$$

where $\gamma_i^{(\mathrm{s})} \in \mathbb{R}$ denotes the scalar-valued subtraction-based normalization term. Injective property allows linear attention (Eq. 3) to distinguish different queries, which is similar to full attention.

## 4 METHOD

### 4.1 LINA MOTIVATION AND ROADMAP

**Motivation.** In this work, we focus on the computation-intensive model side, aiming to design efficient linear attention that accelerates inference while retaining performance. Despite linear attention has shown promise in DiTs (Pu et al., 2024; Xie et al., 2025a;b) and ViTs (Cai et al., 2023; Han et al., 2023; Guo et al., 2024), its suitable design for autoregressive image generation task remains unclear in the literature. To fill the research gap, we highlight three research questions in this work: **Q1** (*linear attention paradigm choice*): which paradigm—division-based or subtraction-based normalization—better supports parameter scaling? **Q2** (*locality choice*): Without softmax, linear attention shows limited ability to model local patterns, which may affect performance. What remedies are effective, and does introducing a locality inductive bias affect parameter scaling? **Q3** (*memory management*): linear attention has fixed memory size to process image tokens without clear differentiation. As tokens vary in semantics, long sequences may cause memory collisions (Yang et al., 2025; Schlag et al., 2021), hindering the focus on informative tokens.

**Roadmap.** Our LINA starts from NOVA (Deng et al., 2025), but safely replaces all full attention with efficient linear attention. As shown in Fig. 2 (with the inference framework detailed in Appendix C), the model consists of: a *Connector* for integrating text or class information; an *Encoder* and *Decoder* for predicting conditioning; and a denoising flow matching network (*i.e.*, a small MLP) (Li et al., 2024b) for modeling token probability distributions. Then, we explore the design of linear attention blocks. First, we conduct a systematic scaling study to compare different linear attention paradigms and the effect of introducing a locality inductive bias through our proposed locality augmentation module (Sec. 4.2, for **Q1**, **Q2**). Then, we introduce the KV Gate as a remedy for memory collisions and empirically evaluate several design choices (Sec. 4.3, for **Q3**).

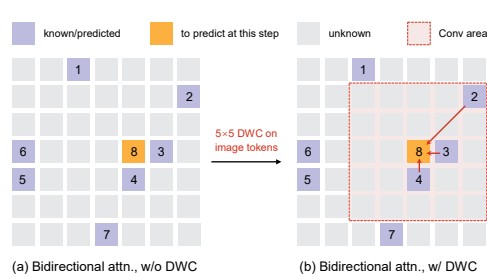

Figure 3: **DWC helps locality.** (a) A random-order autoregressive model with bidirectional attention predicts next tokens based on the predicted tokens. When the target token (*e.g.*, the 8th) is surrounded by predicted tokens (*e.g.*, the 3rd), the model faces challenges due to the limited local modeling capacity. (b) By using convolution layer, DWC module gathers information from nearby known tokens when predicting the current token, thereby facilitating linear attention.

## 4.2 SCALING BEHAVIOR OF LINEAR ATTENTION: AN EMPIRICAL STUDY

### 4.2.1 DESIGN CHOICES 1: LINEAR ATTENTION PARADIGM.

In this section, we empirically compare the scaling behaviors w.r.t. parameter counts of division-based and subtraction-based normalization linear attention (discussed in Sec. 3.2) under the context of autoregressive generation, even though both forms have already been studied in vision perception tasks using ViTs (Dosovitskiy et al., 2020; Touvron et al., 2021; Wang et al., 2021). We consider this comparison to be necessary. Unlike ViTs, which require only one forward pass with all image tokens visible, MAR models generate tokens step by step. At each step, they rely on previously predicted tokens to produce the next set. In the early stages of inference, only a few predicted tokens are available, so effectively exploiting this precious semantic context is vital for generation.

### 4.2.2 DESIGN CHOICES 2: LOCALITY AUGMENTATION.

**`Softmax` may account for the difference in locality modeling between full and linear attention.** A key issue of linear attention is its capacity for local modeling (Han et al., 2023; 2024). Unlike full attention, it does not apply the `softmax` operation. We provide a specific example to conceptually clarify its effect. Consider a matrix $H$ (Eq. 4). After applying `softmax`, the resulting attention matrix $A^{(f)}$ becomes nearly identical to the identity matrix, implying that each token primarily attends to itself—representing the limiting case of local modeling. In this case, although $H$ may differ from the identity matrix, `softmax` helps sharpening the attention distribution, an operation that linear attention lacks. Thus, linear attention may face difficulty in learning local relations.

$$H = \begin{bmatrix} 5.0 & 1.0 & 0.5 \\ 0.5 & 4.0 & 1.0 \\ 1.0 & 0.5 & 3.5 \end{bmatrix}, A^{(f)} = \texttt{softmax}(H) = \begin{bmatrix} 0.9714 & 0.0178 & 0.0108 \\ 0.0280 & 0.9259 & 0.0461 \\ 0.0725 & 0.0440 & 0.8835 \end{bmatrix} \approx \begin{bmatrix} 1 & 0 & 0 \\ 0 & 1 & 0 \\ 0 & 0 & 1 \end{bmatrix}. \quad (4)$$

**Does scaling MAR models benefit from locality augmentation? A conceptual discussion.** As shown in Fig. 3-(a), MAR models generate tokens set by set during inference. When the tokens to be predicted are adjacent to already known tokens, the latter provide valuable context, as neighboring tokens typically exhibit semantic correlations. We argue that, in this context, locality modeling is a crucial property in which linear attention is limited. As shown in Fig. 3-(b), we apply a $5 \times 5$ *depth-wise convolution (DWC)* (Chollet, 2017; Howard et al., 2017) module to potentially cooperate with linear attention, as its effectiveness has been extensively validated in both ViTs (Han et al., 2023; 2024) and DiTs (Wang et al., 2025). During inference, when predicting a token in an autoregressive step, DWC module incorporates neighboring predicted tokens (if available). If these known tokens are spatially close to the target token in 2D space, they tend to share similar semantics, providing useful cues for generation. On the other hand, pure ConvNets show limited scaling potential (*e.g.*, ConvNeXt V2 (Liu et al., 2022; Woo et al., 2023) at ∼650M parameters) compared with vision transformers (*e.g.*, ViT-22B (Dehghani et al., 2023) at ∼22B parameters). As a result, we ablate the use of DWC as a design choice to examine its effect on scaling behavior w.r.t. parameter counts.

**Implementation.** Unlike prior works (Han et al., 2023) that apply convolution to the value states, we add DWC module to image features $I_{\text{img}}$ *only* to the LINA *Decoder*—excluding the query features—and add its output to that of linear attention before the output projection of the attention:

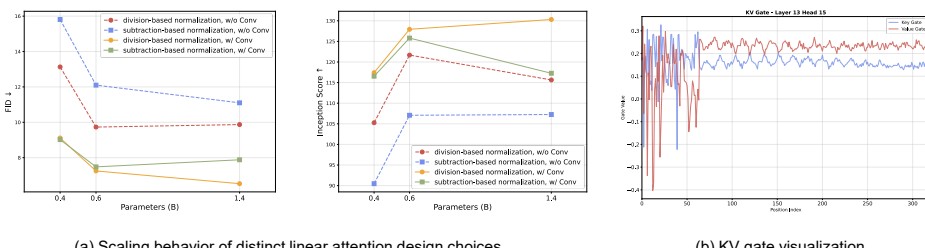

(a) Scaling behavior of distinct linear attention design choices    (b) KV gate visualization

Figure 4: **Scaling behavior and KV gate results.** Fig. (a) describes the class-conditional image generation results on the ImageNet 256×256 benchmark using FID (↓) and IS (↑). Division-based linear attention with LAM achieves the best scaling performance. Detailed results are provided in Appendix H. Fig. (b) presents the learned *KV gate* of a 256px text-to-image LINA model.

$$O^{(\mathrm{d})} = \mathrm{LA}^{(\mathrm{d})}([I_{\mathrm{q}}, I_{\mathrm{img}}]) + \mathrm{DWC}(I_{\mathrm{img}}), \;\; O^{(\mathrm{s})} = \mathrm{LA}^{(\mathrm{s})}([I_{\mathrm{q}}, I_{\mathrm{img}}]) + \mathrm{DWC}(I_{\mathrm{img}}), \tag{5}$$

where $\mathrm{LA}^{(\mathrm{d})}$ and $\mathrm{LA}^{(\mathrm{s})}$ denote linear attention with division-based normalization and subtraction-based normalization, respectively. The reasons are twofold. First, query features $I_{\mathrm{q}}$ already incorporate textual conditioning, rendering the 2D inductive bias of DWC unnecessary. Second, LINA encoder inputs consist only of randomly predicted image tokens, where reshaping them into a 2D layout produces neighborhoods that are not equivalent to those formed by the full set of image features, limiting the benefit of DWC at this stage. With $L_{\mathrm{D}}$ decoder layers, DWC introduces only $k \times k \times D \times L_{\mathrm{D}} \approx 0.31\mathrm{M}$ parameters, which is negligible compared with the overall model size of about 0.4B parameters.

### 4.2.3 SCALING BEHAVIORS

**Experimental setup.** Based on the two design choices discussed above, we have four distinct settings. For each setting, we train models of three sizes, with ∼0.4B, ∼0.6B, and ∼1.4B parameters, respectively. The evaluation is conducted on ImageNet (Deng et al., 2009) class-conditional image generation at 256×256 resolution. All models are trained for 200K iterations on 32 A100 (40GB) GPUs with a learning rate of $8 \times 10^{-4}$. Inference is performed with BFloat16 precision. We report performance without classifier-free guidance (CFG) (Ho & Salimans, 2022), including FID-50K (Heusel et al., 2017), sFID (Nash et al., 2021), Inception Score (Salimans et al., 2016), and Precision/Recall (Kynkäänniemi et al., 2019). Detailed model configuration, hyper-parameters and results are presented in Appendix D, E, and H.

**Division-based normalization scales better than subtraction-based.** Fig. 4 illustrates the scaling performance of the four design choices, reported in terms of FID and IS. We observe that, regardless of the DWC module, division-based linear attention consistently outperforms subtraction-based at the huge scale (∼1.4B). We hypothesize two possible reasons: (1) semantic confusion (Han et al., 2024) may be absent in autoregressive image generation, or its impact is less pronounced than in vision perception models; and (2) at early generation steps, when few tokens are predicted, masked tokens may interfere more with subtraction-based normalization.

**DWC module tends to improve both linear attention variants across model sizes.** From Fig. 4, we further observe that for both forms of linear attention, adding DWC to image features consistently improves performance (*i.e.*, FID and IS) across model sizes. We attribute this to the limited locality of linear attention used, which remains a bottleneck in autoregressive image generation. Introducing an appropriate inductive bias, *e.g.*, depthwise convolution, appears beneficial for parameter scaling. Developing effective ways to enhance locality remains an interesting direction for future work.

*From now on, we will use division-based linear attention with DWC as our basic design choice.*

Table 1: **Ablation study of KV gate.** ImageNet 256×256 results (w/o CFG) are reported. All models are trained for 200K iterations. Head-wise KV gate are chosen.

| Gate | Mode | Key | Value | $z^{(\mathrm{d})}$ | FID ↓ | sFID ↓ | IS ↑ | Pre. ↑ | Rec. ↑ |
|---|---|---|---|---|---|---|---|---|---|
| None | | | | | 9.11 | 5.89 | 117.40 | 0.69 | 0.61 |
| Head-wise | 1 | ✓ | ✓ | | **8.72** | 5.64 | **120.34** | 0.69 | 0.61 |
| Head-wise | 2 | ✓ | | | 8.80 | 5.69 | 119.20 | 0.70 | 0.62 |
| Head-wise | 3 | | ✓ | | 9.06 | 5.60 | 115.90 | 0.69 | 0.61 |
| Head-wise | 4 | ✓ | ✓ | ✓ | 9.22 | 5.89 | 115.41 | 0.69 | 0.61 |
| Head-shared | 1 | ✓ | ✓ | | **8.72** | 5.70 | 120.24 | 0.69 | 0.62 |

## 4.3 KV GATE: FLEXIBLE MEMORY MANAGEMENT IN LINEAR ATTENTION

### 4.3.1 DESIGN PHILOSOPHY

**Why should memory be flexibly regulated in image generative models?** Division-based normalization linear attention (Eq. 2) can be expressed as in the following memory form:

$$M = \sum_{j=1}^{N} \phi(K_j)^{\top} V_j = \sum_{j=1}^{N} M_j, \ \ z = \sum_{m=1}^{N} \phi(K_m)^{\top}, \ \ O_i^{(\mathrm{d})} = \frac{\phi(Q_i)M}{\phi(Q_i)z}, \tag{6}$$

where memory $M \in \mathbb{R}^{D \times D}$ can be interpreted as the *equally weighted sum* of token-wise memories $M_i$, while $z \in \mathbb{R}^{D \times 1}$ contributes to the normalization term $\gamma_i^{(\mathrm{d})} = \phi(Q_i)z$. Modern linear LLMs (Yang et al., 2024; 2025; Sun et al., 2023; 2024b; Dao & Gu, 2024; Schlag et al., 2021) adopt gated linear attention (GLA), which employs a forget gate to discard past information selectively. GLA not only preserves efficiency but also shows remarkable potential in language modeling and commonsense reasoning tasks, *etc.* (Yang et al., 2025). We argue that in autoregressive image generation, memory corresponding to different tokens should be treated suitably, introduced next.

**How to effectively achieve flexible memory arrangement?** *In a nutshell, we use a learnable K gate to scale both $M$ and $z$, and another learnable V gate to scale $M$ only. This method, dubbed KV gate, allows effective and flexible memory arrangement.* Division-based normalization linear attention (Eq. 6) using the *KV gate* can be formulated as follows:

$$\tilde{K}_j = g_j^{(k)} \phi(K_j), \ \ \tilde{V}_j = g_j^{(v)} V_j, \ for \ j \in [1, N]$$

$$M = \sum_{j=1}^{N} \tilde{K}_j^{\top} \tilde{V}_j = \sum_{j=1}^{N} g_j^{(k)} g_j^{(v)} M_j, \ \ z = \sum_{m=1}^{N} \tilde{K}_m^{\top}, \ \ O_i^{(\mathrm{d})} = \frac{\phi(Q_i)M}{\phi(Q_i)z}, \tag{7}$$

where the *K gate* $g^{(k)}$ denotes the scaling coefficient applied to $\phi(K)$ when computing both $M$ and $z$, while the *V gate* $g^{(v)}$ serves as an auxiliary part that adjusts $M$ only.

KV gate is applied in all linear attention in the *Decoder* of our model. Besides the KV gate, we also design three variants as ablations, *i.e.*, *K gate only*, *V gate only*, and a variant with an *extra gate* applied to $z$ (see Appendix E for details of the four modes). Ablation study is conducted on ImageNet class-conditional image generation task using a ∼0.4B model.

As reported in Tab. 1, KV gate consistently improves FID, IS, and sFID compared to division-based normalization linear attention baseline. We present two thought-provoking findings. **(1)** Using only V gate or applying an extra gate to $z$ severely degrades FID and IS. **(2)** Using only K gate slightly affects FID and IS. This indicates that the K gate and V gate exhibit a synergistic effect rather than being mutually exclusive. Moreover, we suggest leveraging the K gate to scale $z$, instead of introducing an additional set of learnable coefficients. With $h$ attention heads, the KV gate introduces only $2 \times h \times N \times L_{\mathrm{D}} \approx 0.12$M parameters for a sequence length of $N = 320$, which is negligible compared to the ∼0.4B parameters of the model.

### 4.3.2 ANALYSIS OF KV GATE

**Should the KV gate be head-wise or head-shared?** We investigate whether the KV gates across different attention heads should share parameters. As shown in Tab. 1, using a head-specific KV gate slightly improves IS and sFID. Given its negligible parameter overhead, we adopt the head-wise design. We kindly note that (Lin et al., 2025; Qiu et al., 2025) investigate incorporating gating mechanisms into full attention. We leave for future work a systematic study of the similarities and differences in how gating influences linear attention *vs.* full attention.

**What pattern has the KV gate learned?** As shown in Fig. 4-(b), we visualize the KV gate of LINA-H trained for 500K iterations at 256px resolution on text-to-image generation task to illustrate specific patterns they have learned. For the first 64 query tokens, the values of the KV gate exhibit fluctuations, while for the subsequent 256 image tokens, they are more stable and display a clear periodic pattern. Notably, K gate peak when V gate dip, indicating complementary roles in memory management in the linear attention. Additional results (provided in Appendix J) show that, although we do not explicitly restrict the KV gate values, they generally remain within the interval $(0, 1)$ and display diverse patterns across different layers and heads. See Appendix J for detailed results.

**Difference between KV gate and the forget gate.** Gated linear attention (Sun et al., 2023; Yang et al., 2024; 2025) in LLMs often introduce a forget gate as a decay term to forget past information, *i.e.*, $M_t = \alpha_t M_{t-1} + K_t^\top V_t$ (assume $\phi(\cdot)$ is the identity). Both the forget gate and our KV gate modulates memory but differs in three ways. **(1)** *Recurrence vs. parallelism:* Forget gate is recursive—its decay factor for $M_j$ is $\prod_{s=j+1}^{t} \alpha_s$. In contrast, our KV gate computes the factor $g_j^{(k)} g_j^{(v)}$ once without recurrence. **(2)** *Data dependence:* Forget gate is typically data-dependent, with $\alpha_t$ projected from the current input token (Yang et al., 2024). Differently, KV gate is data-independent. We find that simple learnable parameters is sufficient to improve performance without linear projection. **(3)** *Range constraint:* Forget gate (Sun et al., 2024b) restricts $\alpha_t$ to $(0, 1)$ due to the `sigmoid` function, whereas KV gate allows $g_j^{(k)}, g_j^{(v)}$ to be negative, supporting flexible learning.

## 5 EXPERIMENTS

### 5.1 CLASS-CONDITIONAL IMAGE GENERATION

**Training details.** As an system-level validation of our method, we conduct class-conditional image generation experiments on the ImageNet 256×256 benchmark using LINA-H (1.4B). Using a learning rate of $8 \times 10^{-4}$ and a batch size of 768, we train the model for a total of 1.2M iterations. We set the model EMA (Polyak & Juditsky, 1992) to 0.99 and the weight decay to 0.02. Following NOVA, we adopt a 6-layer flow matching MLP as the denoising network, without relying on advanced training approaches, *e.g.*, REPA (Yu et al., 2024b). Sampling during inference is done in BFloat16 precision. The autoregressive and diffusion steps are set at 64 and 25, respectively. We report results with CFG scale of 1.0 and 2.4.

Table 2: **ImageNet 256×256 class-conditional image generation.** LINA shows promising results via pure linear attention. Best results are in **bold**; second best are underlined.

| Model | FID ↓ | IS ↑ | Precision ↑ | Recall ↑ |
|---|---|---|---|---|
| *Diffusion models* | | | | |
| ADM (Dhariwal & Nichol, 2021) | 4.59 | 186.70 | 0.82 | 0.52 |
| CDM (Ho et al., 2022) | 4.88 | 158.71 | - | - |
| LDM-4 (Rombach et al., 2022) | 3.60 | 247.67 | 0.87 | 0.48 |
| U-ViT-H/2-G (Bao et al., 2022) | 2.29 | 263.9 | 0.82 | 0.57 |
| DiT-XL/2 (Peebles & Xie, 2023) | 2.27 | 278.24 | 0.83 | 0.57 |
| LiT-XL/2 (Wang et al., 2025) | 2.32 | 265.20 | 0.824 | 0.574 |
| DiffuSSM-XL-G (Yan et al., 2024) | 2.28 | 259.13 | 0.86 | 0.56 |
| DiM-L (Teng et al., 2024) | 2.64 | - | - | - |
| DiM-H (Teng et al., 2024) | 2.21 | - | - | - |
| DiG-XL/2-G Zhu et al. (2024) | 2.07 | 278.95 | 0.82 | 0.60 |
| SiT-XL (Ma et al., 2024) | 2.06 | 277.50 | 0.83 | 0.59 |
| Mediator Pu et al. (2024) | 2.01 | 271.04 | 0.82 | 0.60 |
| *Autoregressive models* | | | | |
| Mask-GIT (Chang et al., 2022) | 6.18 | 182.1 | - | - |
| MAGVIT-v2 (Yu et al., 2024a) | **1.78** | **319.4** | - | - |
| MAR-B (Li et al., 2024b) | 2.31 | 281.7 | 0.82 | 0.57 |
| MAR-L | **1.78** | 296.0 | 0.81 | 0.60 |
| LINA-H (cfg=1.0) | 4.49 | 162.64 | 0.74 | 0.62 |
| LINA-H (cfg=2.4) | 2.18 | 275.73 | 0.81 | 0.58 |

**Results.** In Tab. 2, we present a system-level evaluation of LINA against other frontier models. Compared with the linear diffusion model (*e.g.*, LiT) and the full attention autoregressive model

(*e.g.*, MAR), LINA —based on a linear autoregressive architecture—delivers competitive performance. Notably, LINA-H achieves an FID of 2.18, demonstrating that the linear attention designed in this work is well suited for autoregressive image modeling. In addition, LINA attains superior validated performance than DiM, an efficient state space model (Gu & Dao, 2023; Dao & Gu, 2024) based diffusion model. As a result, we turn to validate LINA on text-to-image benchmarks.

## 5.2 TEXT-TO-IMAGE GENERATION

**Training details.** Our training pipeline consists of three stages. Following prior findings, we initialize Stage 1 with the 1024px NOVA pretrained weights, excluding the linear attention modules. The three stages are trained on 256px, 512px, and 1024px data, respectively. Training is conducted on 48 A100 (40GB) GPUs with batch sizes of 768, 192, and 48. Stages 2 and 3 run for 600K and 700K iterations. We set 128 autoregressive steps and 25 diffusion steps, with a CFG scale of 7.0 during sampling. Further details are provided in Appendix I.

**Results.** As shown in Tab. 4, we compare LINA with advanced text-to-image architectures on the GenEval benchmark (Ghosh et al., 2023). Without prompt engineering, the 1.4B LINA outperforms the 1.6B SANA, an advanced linear DiT baseline. Moreover, compared with autoregressive models using full attention, LINA achieves performance on par with the 10.5B Fluid. These results demonstrate that the proposed linear attention integrates well with autoregressive architectures and exhibits strong text-image alignment, providing a clear and reliable baseline. Fig. 1 shows 1024px samples generated by LINA, where image fidelity and fine textures are well-preserved across long sequences.

Table 3: **Latency comparison results.** LINA competes FlashAttention in 1024px generation.

| Model | Params. | Res. | Type | Acceleration | Latency |
|-------|---------|--------|--------|--------------|---------|
| NOVA | 1.4B | 1024px | Full | FlashAttn | 20.0s |
| LINA | 1.5B | 1024px | Linear | - | 22.0s |

**FLOPs comparison: full attention *vs*. linear attention in LINA** In this section, we compare the FLOPs of linear attention and full attention. We report the FLOPs of *a single* attention module under the following configuration: batch size of 1, sequence length of 5120, hidden dimension of 1536, and 16 attention heads. This setup matches how LINA-H operates at a resolution of 1024px.

The linear attention module we evaluate adopts *division-based normalization* in the LINA *Decoder* and integrates both the *DWC module* and the *KV gate* proposed in this work. FLOPs are measured using the fvcore (fvcore Contributors, 2021) library.

The results are presented in Fig. 5. A full-attention module requires approximately ∼129 GFLOPs, whereas the linear-attention module requires only ∼50 GFLOPs. This corresponds to a reduction of about ∼61% in computation, highlighting the efficiency benefits of our LINA. Importantly, despite this substantial reduction in FLOPs, the text-to-image performance of LINA remains competitive with the full-attention-based NOVA (Deng et al., 2025).

For clarity, we note that the *DWC module* and the *KV gate* are used only in the LINA *Decoder*, and are not applied in the *Encoder* or the *Connector*.

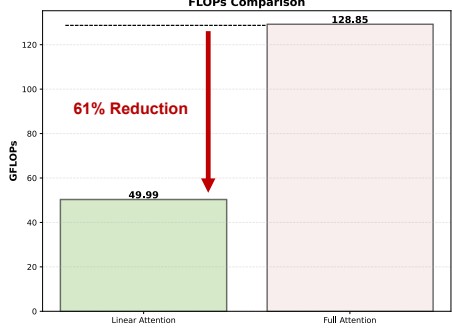

Figure 5: **FLOPs comparison results.** We compare *a single* module of linear attention and full attention under the following setting: batch size 1, sequence length 5120, hidden dimension 1536, and 16 heads. This configuration corresponds to how LINA-H operates at 1024px resolution. The linear attention variant applies division-based normalization and incorporates both the *DWC module* and the *KV gate*, explored heavily in this work. Compared with the full attention, *a single* linear attention module reduces FLOPs by ∼61%, showing computation efficiency.

**Latency comparison: FlashAttention *vs*. LINA.** Tab. 3 reports the pipeline latency comparison between NOVA (Deng et al., 2025) and LINA when generating 1024px images. LINA shares the

Table 4: **Comparison of GenEval results.** Rewriter refers to a prompt engineering method (Deng et al., 2025). Our LINA, equipped with pure linear attention, rivals advanced T2I frameworks. Best results are in **bold**; second best are underlined.

| Model | Params. | Overall | Single | Two | Counting | Colors | Position | ColorAttr |
|---|---|---|---|---|---|---|---|---|
| *Diffusion models* | | | | | | | | |
| PixArt-$\alpha$ (Chen et al., 2024b) | 0.6B | 0.48 | 0.98 | 0.50 | 0.44 | 0.80 | 0.08 | 0.07 |
| LiT (1024×1024) (Wang et al., 2025) | 0.6B | 0.48 | 0.98 | 0.50 | 0.40 | 0.77 | 0.11 | 0.12 |
| LiT (512×512) | 0.6B | 0.47 | 0.97 | 0.43 | 0.42 | 0.79 | 0.09 | 0.12 |
| DALL-E3 (OpenAI, 2023) | - | 0.67 | 0.96 | 0.87 | 0.47 | 0.83 | 0.43 | 0.45 |
| SDXL (Podell et al., 2023) | 2.6B | 0.55 | 0.98 | 0.44 | 0.39 | 0.85 | 0.15 | 0.23 |
| SD3 (Esser et al., 2024) | 2B | 0.62 | 0.98 | 0.74 | 0.63 | 0.67 | 0.34 | 0.36 |
| Playground v2.5 (Li et al., 2024a) | 2.6B | 0.56 | - | - | - | - | - | - |
| Hunyuan-DiT (Li et al., 2024c) | 1.5B | 0.63 | - | - | - | - | - | - |
| SANA (1024×1024) (Xie et al., 2025a) | 1.6B | 0.66 | - | - | - | - | - | - |
| SANA (512×512) | 1.6B | 0.66 | - | - | - | - | - | - |
| *Autoregressive models* | | | | | | | | |
| LlamaGen (Sun et al., 2024a) | 0.8B | 0.32 | 0.71 | 0.34 | 0.21 | 0.58 | 0.07 | 0.04 |
| Emu3 (+ Rewriter) (Wang et al., 2024) | 8B | 0.66 | 0.99 | 0.81 | 0.42 | 0.80 | 0.49 | 0.45 |
| Show-o (Xie et al., 2024) | 1.3B | 0.53 | 0.95 | 0.52 | 0.49 | 0.82 | 0.11 | 0.28 |
| NOVA (1024×1024) (Deng et al., 2025) | 1.4B | 0.71 | 0.99 | 0.91 | 0.62 | 0.85 | 0.33 | 0.56 |
| NOVA (512×512) (+ Rewriter) | 0.6B | **0.75** | 0.98 | 0.88 | 0.62 | 0.82 | 0.62 | 0.58 |
| Fluid (Fan et al., 2025a) | 1.1B | 0.67 | 0.96 | 0.77 | 0.61 | 0.78 | 0.34 | 0.53 |
| Fluid (Fan et al., 2025a) | 10.5B | 0.69 | 0.96 | 0.83 | 0.63 | 0.80 | 0.39 | 0.51 |
| LINA-H (1024×1024) | 1.5B | 0.66 | 0.99 | 0.85 | 0.50 | 0.88 | 0.38 | 0.39 |
| + Rewriter | 1.5B | 0.72 | 0.99 | 0.84 | 0.54 | 0.85 | 0.56 | 0.53 |
| LINA-H (512×512) | 1.4B | 0.68 | 0.98 | 0.83 | 0.56 | 0.89 | 0.34 | 0.50 |
| + Rewriter | 1.4B | 0.74 | 0.99 | 0.85 | 0.61 | 0.87 | 0.60 | 0.53 |

same macro architecture as NOVA (Deng et al., 2025), but replaces NOVA's full attention with linear attention throughout. For each run, we compute the average latency of generating 10 images, and the reported latency is the mean over three such runs. All experiments are conducted on a single NVIDIA A800 GPU with a batch size of 1. Since our work represents an early exploration of linear MAR models with an emphasis on sample quality, we do not incorporate additional acceleration techniques, *e.g*., Triton (Tillet et al., 2019). Even without such optimizations, LINA attains latency on par with full attention using FlashAttention (Dao et al., 2022). We attribute this parity to the inherent computational advantage of linear attention in processing long sequences. Our design serves as a building block for linear MAR, readily complementing future acceleration methods.

## 6  CONCLUSION

This paper takes a deep dive into how linear attention should be designed for autoregressive image generation with continuous tokens. Through a comprehensive empirical study, we recommend adopting division-based normalization and incorporating convolution to strengthen locality. Besides, we introduce the KV gate, a simple mechanism that modulates key and value states to enable flexible memory management and, in turn, improve generation. Our final model, dubbed LINA, is an linear autoregressive model that runs fast and delivers promising image generation performance. We envision this work as a building block that paves the way for developing acceleration techniques in linear attention toward real-world applications, which we leave for future work.

## REPRODUCIBILITY STATEMENT

To ensure reproducibility, we have made deliberate efforts.

As our main contribution lies in architectural exploration, we provide the model code files in the supplementary materials for readers' reference.

Details of the training procedure are described in Sec. 5.1 and Sec. 5.2 of the main paper, while the experimental hyperparameters are thoroughly documented in Appendix E and Appendix I.

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

## A    THE USE OF LARGE LANGUAGE MODELS (LLMS)

To ensure accuracy in language and faithfully convey the intended ideas, this paper was prepared with the assistance of large language models (LLMs). While not all generated outputs were adopted, LLMs directly or indirectly contributed to the writing process. In particular, the LLM was used for: 1) Refining language expression by making the author's draft more concise, natural, and clear. 2) Assessing whether the text was idiomatic and precise, and suggesting potential revisions. 3) Checking for possible grammatical errors and offering corrections. 4) Assisting in parts of the experimental code. The author remains fully responsible for the authenticity and integrity of all content in this paper.

## B    FULL RELATED WORK

**Efficient image generation.**    Efficiency is a key concern for image generative models when dealing with long sequences.

Research on efficient diffusion models is relatively extensive. Some approaches focus on architectural modifications, especially efficient attention (Katharopoulos et al., 2020; Choromanski et al., 2021; Yang et al., 2024; Cai et al., 2023; Han et al., 2023). For example, SANA (Xie et al., 2025a) studies the text encoder in text-to-image models, while LiT (Wang et al., 2025) provides guidelines for converting a pretrained DiT into a linear DiT. DiG (Zhu et al., 2024) and DiM (Teng et al., 2024) explore applying gated linear attention and state space models (Gu & Dao, 2023; Dao & Gu, 2024), respectively, to image generation. Other works pursue fewer or even single diffusion steps, such as DMD (Yin et al., 2024b), DMD2 (Yin et al., 2024a), CausVid (Yin et al., 2025), consistency models (Song et al., 2023; Lu & Song, 2024), and SANA-Sprint (Chen et al., 2025).

For autoregressive models with continuous tokens, researchers have explored various directions to improve efficiency. Notably, DiSA (Zhao et al., 2025) reduces the number of diffusion steps as the autoregressive process progresses. LazyMAR (Yan et al., 2025) explores how to use feature caching to improve efficiency while maintaining performance. DC-AR (Wu et al., 2025) studies the design and training of image tokenizer. ARFlow (Hui et al., 2025) enables flow-based image generation through hybrid linear attention.

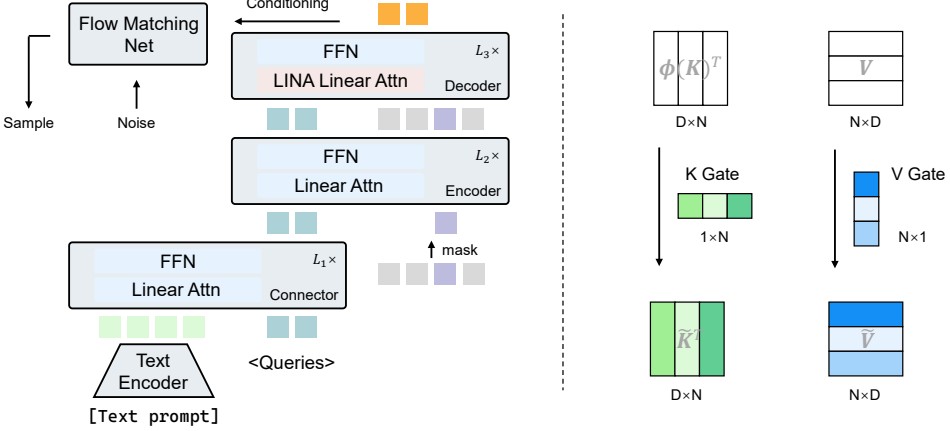

(a) Our LINA generation framework for inference

(b) KV gate for flexible memory management

Figure 6: **Overview of LINA**: Fig. (a) illustrates the inference pipeline. LINA builds its Connector, Encoder, and Decoder entirely with linear attention in pursuit of efficiency. At each step, the Encoder and Decoder predict the conditioning from the known tokens, after which the denoising network draws sample based on the conditioning.

## C    INFERENCE PIPELINE

As described in Fig. 6-(a), LINA inference pipeline starts by encoding the text prompt with a text encoder and integrating it into the query token through the Connector. The image generation process starts with all tokens masked and proceeds through a multi-step, generalized autoregressive procedure that generates image tokens progressively. At each step, the Encoder extracts information from the predicted tokens, which—together with the masked tokens—are decoded by the Decoder to form the conditioning. A denoising flow matching network (*e.g.*, MLP) samples the token based on this conditioning. After all autoregressive steps, the generated tokens are feed into a VAE decoder to produce the final image.

## D    MODEL CONFIGURATION

The detailed configurations of our LINA models of varying sizes for class-conditional image generation are listed in Tab. 5, corresponding to the exploration roadmap discussed in Sec. 4.2. Tab. 6 reports the detailed hyperparameters of LINA for both class-conditional and text-to-image generation in our main experiments in Sec. 5.

Table 5: Detailed LINA configurations in Sec. 4.2.

| Configuration | L | XL | H |
|---|---|---|---|
| Task | C2I | C2I | C2I |
| Resolution | 256px | 256px | 256px |
| Params (M) | 0.4B | 0.6B | 1.4B |
| Connector Blocks | 16 | 16 | 16 |
| Encoder Blocks | 16 | 16 | 16 |
| Decoder Blocks | 16 | 16 | 16 |
| Flow Matching MLP Depth | 6 | 6 | 6 |
| Channels | 768 | 1024 | 1536 |
| DWC Kernel Size | 5 | 5 | 5 |

Table 6: Detailed LINA configurations in Sec. 5.

| Configuration | H | H | H |
|---|---|---|---|
| Task | C2I | T2I | T2I |
| Resolution | 256px | 512px | 1024px |
| Params (M) | 1.4B | 1.4B | 1.5B |
| Connector Blocks | 16 | 16 | 16 |
| Encoder Blocks | 16 | 16 | 16 |
| Decoder Blocks | 16 | 16 | 16 |
| Flow Matching MLP Depth | 6 | 6 | 6 |
| Channels | 1536 | 1536 | 1536 |
| DWC Kernel Size | 5 | 5 | 5 |

Table 7: Training setting of LINA for scaling behavior empirical study in Sec. 4.2.

| Training Setting | L | XL | H |
|---|---|---|---|
| Base Learning Rate | $8 \times 10^{-4}$ | $8 \times 10^{-4}$ | $8 \times 10^{-4}$ |
| Batch Size | 64×32 | 48×32 | 24×32 |
| Training Iteration | 200K | 200K | 200K |
| Weight Decay | 0.02 | 0.02 | 0.02 |
| Warm-up Steps | 10000 | 10000 | 10000 |
| Model EMA | 0.99 | 0.99 | 0.99 |

## E    Detailed hyper-parameters on ImageNet-1K in Sec. 4.2

In Tab. 7, we provide the hyper-parameters for the experiments in Sec. 4.2, which explore the scaling behavior of different linear attention design choices.

## F    Complexity analysis of DWC Module and KV Gate

We follow the notation in Section 2.2: assume the model uses $H$ attention heads to process $N = N_q + N_{img}$ tokens, where each token has a hidden dimension $D$, and the per-head dimension is $d$, satisfying $D = hd$.

**Linear attention.**    The computation of the per-token output $O_i^{(d)}$ for linear attention with division-based normalization can be expressed as:

$$O_i^{(d)} = \frac{\phi(Q_i)\left(\sum_{j=1}^{N}\phi(K_j)^{\top}V_j\right)}{\phi(Q_i)\left(\sum_{m=1}^{N}\phi(K_m)^{\top}\right)},$$
(8)

Theoretically, for the numerator, we have:

$$\underbrace{\phi(Q_i)}_{\mathbb{R}^{1\times d}}\left(\sum_{j=1}^{N}\underbrace{\phi(K_j)^{\top}}_{\mathbb{R}^{d\times 1}}\underbrace{V_j}_{\mathbb{R}^{1\times d}}\right) \longrightarrow \mathbb{R}^{1\times d},$$
(9)

where, first, a $d \times 1$ matrix is multiplied by a $1 \times d$ matrix, which costs $\mathcal{O}(d^2)$. Then, a $1 \times d$ matrix is multiplied by an a $d \times d$ matrix, which also costs $\mathcal{O}(d^2)$.

For the denominator, we have:

$$\underbrace{\phi(Q_i)}_{\mathbb{R}^{1\times d}}\left(\sum_{m=1}^{N}\underbrace{\phi(K_m)^{\top}}_{\mathbb{R}^{d\times 1}}\right) \longrightarrow \mathbb{R}^{1\times 1},$$
(10)

where, a $1 \times d$ matrix is multiplied by a $d \times 1$ matrix, which costs $\mathcal{O}(d)$, which is negligible in the big-$\mathcal{O}$ sense.

Computing the final output $O_i^{(d)}$ also costs $\mathcal{O}(d)$, which is negligible in the big-$\mathcal{O}$ sense. Thus, the per-token complexity is at most $\mathcal{C}_{la,t} = 2d^2 = \mathcal{O}(d^2)$.

For a single attention head processing $N$ tokens, the total cost is at most:

$$\text{Linear attention, per-head:} \quad \mathcal{C}_{la,h} = N\mathcal{C}_{la,t} = \mathcal{O}(Nd^2).$$
(11)

For the whole linear attention with $h$ heads (and hidden dimension $D = hd$), the total cost is:

$$\text{Linear attention:} \quad \mathcal{C}_{la} = h\mathcal{C}_{la,h} = \mathcal{O}(NDd).$$
(12)

**Depthwise convolution.**    The DWC module applies a $k \times k$ depthwise convolution to $N_{img}$ image tokens only, with a cost of:

$$\text{DWC module:} \quad \mathcal{C}_{dwc} = \mathcal{O}(N_{img}Dk^2).$$
(13)

The ratio between the computational cost of the *DWC module* and that of *linear attention* is:

$$\frac{\mathcal{O}(N_{img}Dk^2)}{\mathcal{O}(NDd)} \leq \frac{k^2}{d}.$$
(14)

Note that the DWC module processes only $N_{\text{img}}$ image tokens, while the linear attention processes both $N_{\text{q}}$ query tokens and $N_{\text{img}}$ image tokens ($N = N_{\text{q}} + N_{\text{img}}$).

For our text-to-image model LINA-H, we have $D = 1536$ and $h = 16$, so the per-head dimension is $d = 96$. Therefore, we obtain $k^2/d = 0.26$.

Note that the actual cost ratio is **smaller** than $0.26$. The reasons are as follows. The table below provides detailed configurations of LINA. As shown, the DWC module is used only in the *Decoder* blocks, and these Decoder blocks constitute *only $\frac{1}{3}$ of the total number of blocks*. Therefore, the DWC module introduces only non-dominant additional computational cost.

**KV gate.** The KV gate $g^{(k)}, g^{(v)} \in \mathbb{R}^{N_{\text{img}}}$ uses learnable parameters to scale the keys and values on a per-token basis.

$$\tilde{K}_j = \underbrace{g_j^{(k)}}_{\mathbb{R}^{1 \times 1}} \underbrace{\phi(K_j)}_{\mathbb{R}^{1 \times d}}, \quad \tilde{V}_j = \underbrace{g_j^{(v)}}_{\mathbb{R}^{1 \times 1}} \underbrace{V_j}_{\mathbb{R}^{1 \times d}}, \tag{15}$$

For a single head processing $N_{\text{img}}$ tokens, the cost is at most:

$$\text{KV gate, per-head:} \quad \mathcal{C}_{kvg,h} = 2N_{\text{img}}d = \mathcal{O}(N_{\text{img}}d). \tag{16}$$

For the whole linear attention with $h$ heads (and hidden dimension $D = hd$), the total cost is:

$$\text{KV gate:} \quad \mathcal{C}_{kvg} = h\mathcal{C}_{kvg,h} = \mathcal{O}(N_{\text{img}}D). \tag{17}$$

The ratio between the computational cost of the *KV gate* and that of *linear attention* is:

$$\frac{\mathcal{O}(N_{\text{img}}D)}{\mathcal{O}(NDd)} \leq \frac{1}{d}. \tag{18}$$

Note that the KV gate processes only $N_{\text{img}}$ image tokens, while the linear attention processes both $N_{\text{q}}$ query tokens and $N_{\text{img}}$ image tokens ($N = N_{\text{q}} + N_{\text{img}}$).

Since the per-head dimension is $d = 96$, the additional computational cost introduced by the KV gate is negligible.

**Comparison to full attention.** The computational complexity of standard softmax attention is $\mathcal{C}_{fa} = O(N^2 D)$, and the ratio between linear attention and full attention is:

$$\frac{\mathcal{C}_{la}}{\mathcal{C}_{fa}} = \frac{\mathcal{O}(NDd)}{\mathcal{O}(N^2D)} = \frac{d}{N}. \tag{19}$$

Note that when LINA operates at 1024px, we have $N = 5120$, $D = 1536$, and $h = 16$, so $d = D/h = 96$. This gives a complexity ratio of $d/N = 96/5120 \approx 0.019$. Therefore, linear attention achieves a substantial reduction in computational cost, which is consistent with our measured FLOPs.

Table 8: Detailed configurations for LINA-H. The DWC module and the KV gate are used only in the LINA *Decoder*, introducing only a small amount of additional parameters and computation.

| Structure | Connector | Encoder | Decoder |
|---|---|---|---|
| Block Number | 16 | 16 | 16 |
| Block Number | 16 | 16 | 16 |
| Block Number | 16 | 16 | 16 |
| Channels | 1536 | 1536 | 1536 |
| DWC Module | ✗ | ✗ | ✓ |
| KV Gate | ✗ | ✗ | ✓ |

## G   How Our Findings Relate to Prior Work on Linear Attention

We note that neither the normalization types used in linear attention nor the locality augmentation techniques are our original inventions. However, to the best of our knowledge, our work is the *first* to systematically investigate these components in the context of *autoregressive image generation*. Prior studies such as Flattened Transformer Han et al. (2023) and InLine Han et al. (2024) only explored them in perception tasks such as image classification.

To clarify the relationship between our findings and existing results in other domains, we summarize the key similarities and differences below.

**Differences from visual perception tasks.**   In autoregressive image generation, our results show that *division-based normalization* yields better *scaling behavior* for linear attention than subtraction-based normalization. This observation is not fully aligned with the conclusions drawn from perception tasks (*e.g.*, InLine). We suspect that this discrepancy may arise from the inherent gap between generation and perception tasks. Issues such as "semantic confusion", highlighted by InLine, may not be the main bottleneck in generative models. Instead, division-based normalization might be inherently more suitable for denoising-based generative processes. We admit that we do not yet have a theoretical explanation for this phenomenon. Nonetheless, we believe that the empirical results and the new data points we provide will be valuable for guiding future work.

**Similarities to visual perception tasks.**   Consistent with findings in perception tasks (*e.g.*, Flattened Transformer), our autoregressive generation results confirm that *linear attention indeed benefits from additional locality modeling*. This can be seen in the scaling behavior in Fig. 4-(a) in the main paper. Nevertheless, our work further raises two conceptual questions in Sec. 4.2, *i.e.*, 1) Whether the softmax operation is the key reason behind the locality gap between full attention and linear attention; and 2) Whether locality modeling universally helps autoregressive image generation. Although our conclusions overlap with prior work in image classification, we believe these discussions offer useful conceptual insights into the role of locality in linear attention for autoregressive generation.

In summary, our results suggest that *linear attention in autoregressive image generation comes with task-specific considerations*, such as preferring division-based normalization, rather than directly inheriting conclusions from perception tasks. We hope our study can provide reliable guidelines for future research, helping to avoid large amounts of repetitive ablations.

## H   Scaling Behavior: Detailed Results

In Tab. 9, we present detailed results, as discussed in Sec. 4.2, comparing the scaling behavior of the four linear attention design choices. For evaluation metrics, we literally reported include FID-50K (Heusel et al., 2017), sFID (Nash et al., 2021), Inception Score (Salimans et al., 2016), and Precision/Recall (Kynkäänniemi et al., 2019).

## I   Detailed hyper-parameters on Text-to-image Generation

In Tab. 10, we present the training setting for the text-to-image generation experiments in Sec. 5, including the dataset size, batch size, learning rate, *etc.*. The training process can be divided into three stages. Our training is conducted using 48 NVIDIA A100 (40G) GPUs.

## J   KV Gate

### J.1   Ablation of KV gate Designs

The variants of the KV gate ablation are listed below.

Table 9: Scaling performance of different linear attention design choices. We report the results on the class-conditional image generation using the ImageNet benchmark at $256 \times 256$ resolution.

| Linear Attention Setting | FID↓ (w/o cfg) | sFID↓ | IS↑ | Precision↑ | Recall↑ |
|---|---|---|---|---|---|
| *Large (L)*: | | | | | |
| Division-based Normalization, w/o DWC | 13.13 | 7.58 | 105.26 | 0.65 | 0.61 |
| Subtraction-based Normalization, w/o DWC | 15.81 | 8.95 | 90.47 | 0.62 | 0.60 |
| Division-based Normalization, w/ DWC | 9.11 | 5.89 | 117.40 | 0.69 | 0.61 |
| Subtraction-based Normalization, w/ DWC | 9.02 | 5.77 | 116.54 | 0.70 | 0.61 |
| *Extra Large (XL)*: | | | | | |
| Division-based Normalization, w/o DWC | 9.73 | 6.46 | 121.68 | 0.68 | 0.60 |
| Subtraction-based Normalization, w/o DWC | 12.10 | 7.50 | 107.07 | 0.66 | 0.60 |
| Division-based Normalization, w/ DWC | 7.25 | 5.17 | 127.93 | 0.71 | 0.61 |
| Subtraction-based Normalization, w/ DWC | 7.48 | 5.38 | 125.80 | 0.72 | 0.60 |
| *Huge (H)*: | | | | | |
| Division-based Normalization, w/o DWC | 9.87 | 6.05 | 115.64 | 0.69 | 0.58 |
| Subtraction-based Normalization, w/o DWC | 11.10 | 7.08 | 107.25 | 0.67 | 0.59 |
| Division-based Normalization, w/ DWC | 6.53 | 5.01 | 130.31 | 0.72 | 0.60 |
| Subtraction-based Normalization, w/ DWC | 7.88 | 5.28 | 117.26 | 0.72 | 0.58 |

Table 10: Training setting of our text-to-image generation LINA.

| Training Setting | Stage 1 | Stage 2 | Stage 3 |
|---|---|---|---|
| Training Iterations | 565K | 600K | 50K |
| Dataset Size | 28M | 28M | 16M |
| Resolution | 256px | 512px | 1024px |
| Base Learning Rate | $2e^{-4}$ | $1e^{-4}$ | $5e^{-5}$ |
| Batch Size | $16 \times 48$ | $4 \times 48$ | $1 \times 48$ |
| Weight Decay | 0.02 | 0.02 | 0.02 |
| Warm-up Steps | 10000 | 10000 | 10000 |
| Model EMA | 0.99 | 0.99 | 0.99 |

**Mode 1 (*KV gate*).**

$$\tilde{K}_j = g_j^{(k)}\phi(K_j), \ \ \tilde{V}_j = g_j^{(v)}V_j, \ for \ j \in [1, N]$$

$$M = \sum_{j=1}^N \tilde{K}_j^\top \tilde{V}_j = \sum_{j=1}^N g_j^{(k)} g_j^{(v)} M_j, \ \ z = \sum_{m=1}^N \tilde{K}_m^\top, \ \ O_i^{(\mathrm{d})} = \frac{\phi(Q_i)M}{\phi(Q_i)z}. \tag{20}$$

where $g^{(k)}, g^{(v)} \in \mathbb{R}^N$ are learnable parameters.

**Mode 2 (*K gate*).**

$$\tilde{K}_j = g_j^{(k)}\phi(K_j), \ for \ j \in [1, N]$$

$$M = \sum_{j=1}^N \tilde{K}_j^\top V_j = \sum_{j=1}^N g_j^{(k)} M_j, \ \ z = \sum_{m=1}^N \tilde{K}_m^\top, \ \ O_i^{(\mathrm{d})} = \frac{\phi(Q_i)M}{\phi(Q_i)z}. \tag{21}$$

where $g^{(k)} \in \mathbb{R}^N$ are learnable parameters.

**Mode 3 (*V gate*).**

$$\tilde{V}_j = g_j^{(v)}V_j, \ for \ j \in [1, N]$$

$$M = \sum_{j=1}^N \phi(K_j)^\top \tilde{V}_j = \sum_{j=1}^N g_j^{(v)} M_j, \ \ z = \sum_{m=1}^N \phi(K_m)^\top, \ \ O_i^{(\mathrm{d})} = \frac{\phi(Q_i)M}{\phi(Q_i)z}. \tag{22}$$

where $g^{(v)} \in \mathbb{R}^N$ are learnable parameters.

**Mode 4 (*KV gate + extra $z$ gate*).**

$$\tilde{K}_j = g_j^{(k)}\phi(K_j), \ \ \bar{K}_j = g_j^{(n)}\phi(K_j), \ \ \tilde{V}_j = g_j^{(v)}V_j, \ for \ j \in [1, N]$$

$$M = \sum_{j=1}^N \tilde{K}_j^\top \tilde{V}_j = \sum_{j=1}^N g_j^{(k)} g_j^{(v)} M_j, \ \ z = \sum_{m=1}^N \bar{K}_m^\top, \ \ O_i^{(\mathrm{d})} = \frac{\phi(Q_i)M}{\phi(Q_i)z}. \tag{23}$$

where $g^{(k)}, g^{(v)}, g^{(n)} \in \mathbb{R}^N$ are learnable parameters.

### J.2 VISUALIZATION

Figure 7 presents detailed visualizations of the learned KV gate values across different layers and heads. From the visualizations, we find a common pattern across layers and heads: the first 64 query tokens show fluctuations, while the following 256 image tokens remain relatively stable with distinct behaviors. We leave the analysis of text tokens to future work and focus here on the KV gate patterns for image tokens.

Our observations are three-folds: (1) From a cross-layer perspective, the KV gate patterns for image tokens also differ by layer. For example, in layer 1, head 1 the fluctuations are more pronounced, indicating substantial variation in KV gate values across image tokens. In contrast, in layer 13, head 1 the fluctuations are much weaker, suggesting that the values remain relatively stable across tokens. (2) From the perspective of value ranges, in many layers the KV gate values lie mostly within $(-1, 1)$, such as layer 10, head 6, layer 4, head 1, and layer 4, head 11. This suggests that, in certain cases, the model applies token-wise attenuation to both $M$ and $z$ when regulating memory in linear attention. (3) From the perspective of heads, the KV gate within the same layer seems to exhibit diverse patterns across different heads. For example, in layer 16, the fluctuations of head 6 and head 11 seems relatively stable, whereas head 1 seems to oscillate more strongly.

## K MORE QUALITATIVE RESULTS

We provide additional qualitative results sampled from 1024px images generated by LINA. Fig. 8 and 9 correspond to shorter text prompts, while Fig. 10 corresponds to a longer prompt. LINA produces high-fidelity images with convincing details and textures. These results support our belief that the proposed LINA offers both practical effectiveness and potential value for generative modeling.

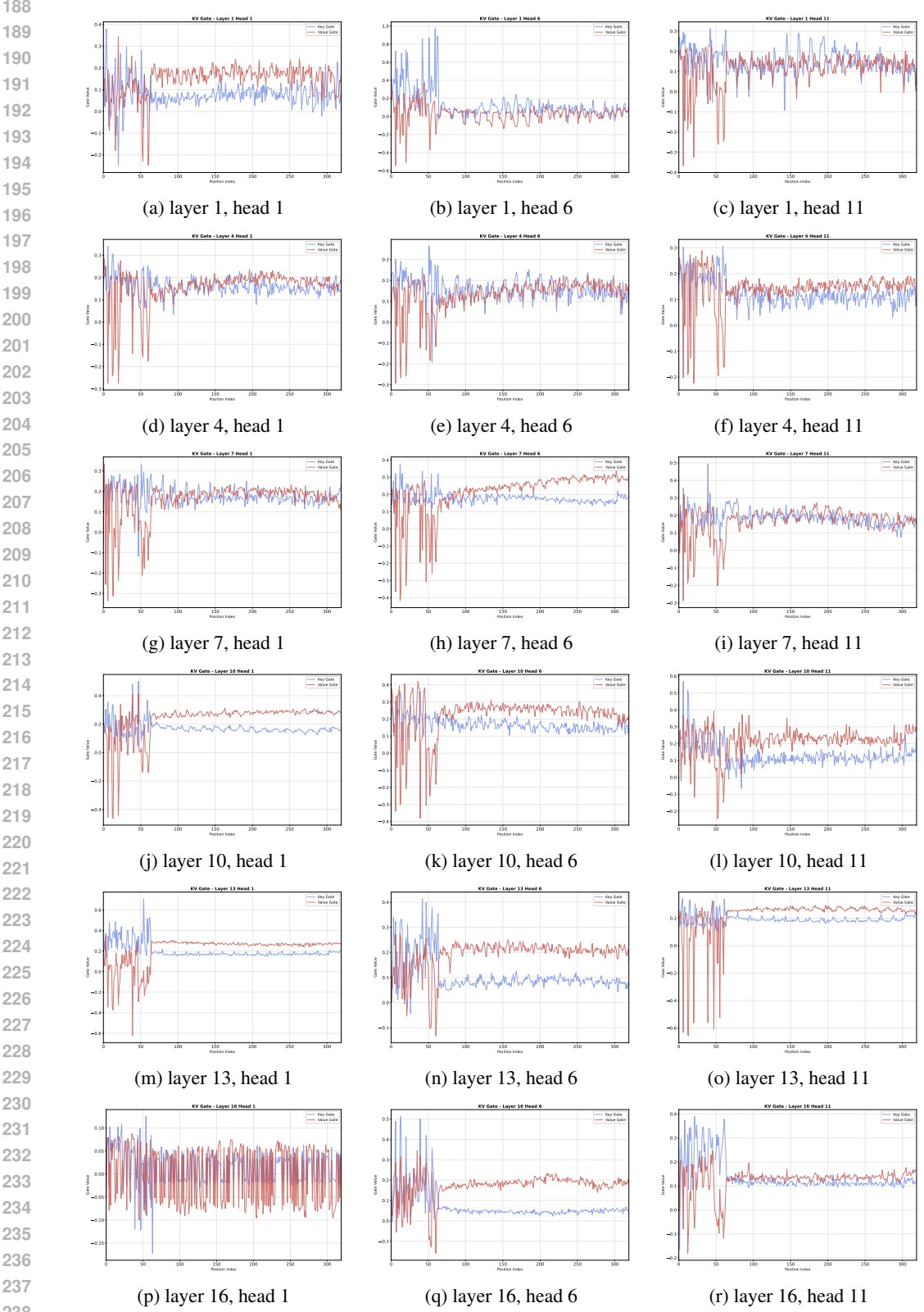

Figure 7: **KV gate visualization.** We plot the *KV gate* results for layers 1, 4, 7, 10, 13, 16 (indexed 1–16) and heads 1, 6, 11 (indexed 0–15). Across different layers and heads, the *KV gate* learns distinct patterns, allowing flexible memory management. Zoom in for best view.

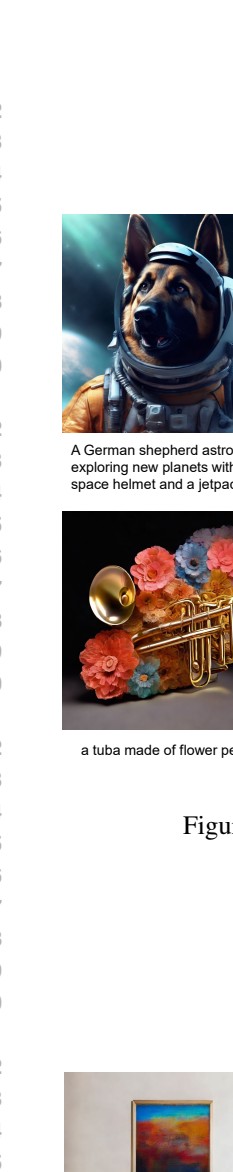
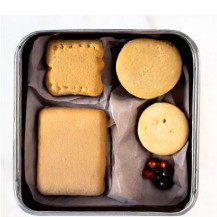
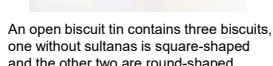
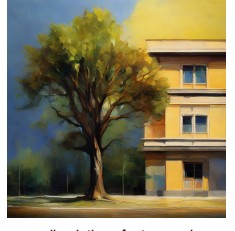
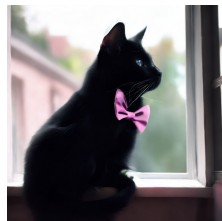

A German shepherd astronaut exploring new planets with a space helmet and a jetpack.

An open biscuit tin contains three biscuits, one without sultanas is square-shaped and the other two are round-shaped.

an oil painting of a tree and a building

A black cat wearing a pink bow tie sits on the windowsill and stares into the distance.

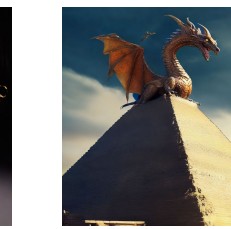
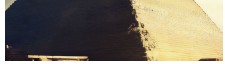
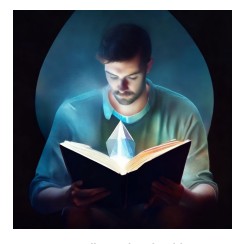
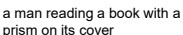
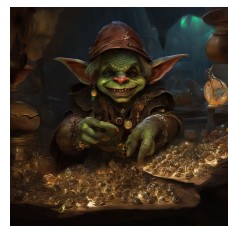

a tuba made of flower petals

a dragon perched on top of the Great Pyramid

a man reading a book with a prism on its cover

A goblin trading shiny trinkets in a hidden, mystical market.

Figure 8: Detailed qualitative results: 1024px samples from LINA, Part 1.

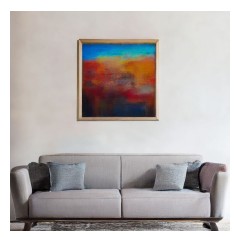
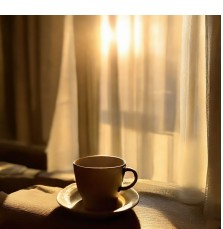
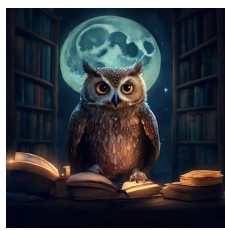
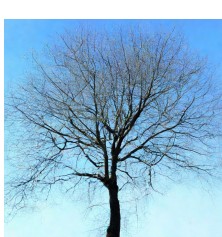

A painting centered above a sofa, slightly to the left.

Sunlight filtering through the linen curtains, casting a warm glow on the ceramic coffee mug.

An owl with a tiny book in a moonlit library.

a summer tree without any leaves

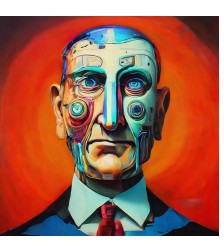
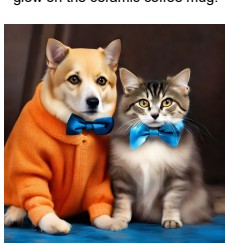
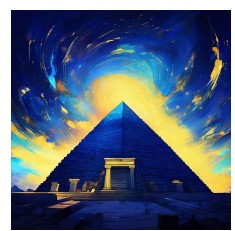
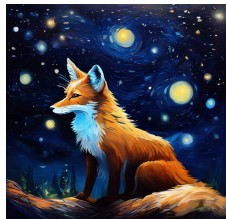

vibrant portrait painting of Salvador Dalí with a robotic half face

A small dog in a cozy orange sweater sitting beside a cat wearing a stylish blue bow tie.

Anime illustration of the Great Pyramid sitting next to the Parthenon under a blue night sky of roiling energy, exploding yellow stars, and chromatic blue swirls

a painting of a fox in the style of starry night

Figure 9: Detailed qualitative results: 1024px samples from LINA, Part 2.

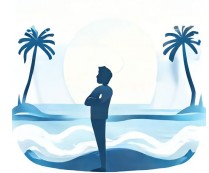

Drone view of waves crashing against the rugged cliffs along Big Sur's garay point beach. The crashing blue waters create white-tipped waves, while the golden light of the setting sun illuminates the rocky shore. A small island with a lighthouse sits in the distance, and green shrubbery covers the cliff's edge. The steep drop from the road down to the beach is a dramatic feat, with the cliff's edges jutting out over the sea. This is a view that captures the raw beauty of the coast and the rugged landscape of the Pacific Coast Highway.

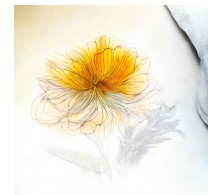

a stylized, cartoon-like illustration. In the foreground, there's a silhouette of a person standing with their arms crossed, facing towards the right side of the image. This central figure is depicted in a dark blue color, outlined against a lighter blue and white background. In the background, there are two palm trees on either side, bending slightly towards the center. Between the palm trees, a sun or moon can be seen partially obscured by them on a light blue backdrop, suggesting it could be either sunrise or sunset. Below, there are stylized representations of waves, indicating the presence of water, possibly symbolizing an ocean or sea. The overall lighting in the image is flat with no shadows or highlights, which aligns with the graphic, two-dimensional style of the art. The person in the image does not show any discernible facial expression due to the silhouette style, and there's no visible text present in the image. The color palette used gives the image a calm and cool feel, with different shades of blue dominating the composition.

an abstract design resembling a flower with a central radiating motif, constructed from fine lines and tones. The flower itself appears to be a marigold or similar flower, depicted with a high level of detail and an artistic interpretation that uses layered line work to create depth and dimension. The design has a gradient of color from the center of the flower, where it's a bright yellow-orange, to the petals' edges, which are rendered in lighter shades. At the base of the floral design, there are structures that resemble fern leaves or similar foliage which are intricately detailed, adding to the overall flora-inspired theme. The background of the image seems to be a textured paper or a surface that mimics such, with the right-hand side being brighter, suggesting some light source out of frame to the right. The fine lines that make up both the flower and the foliage have slight variations in thickness, implying a hand-drawn or digitally-stylized technique. The overall style is reminiscent of pen and ink drawings with a touch of watercolor or digital painting techniques. There is no discernible text in the image, allowing the viewer to focus solely on the art. The image is not in a photo-realistic style but rather an artistic representation that emphasizes the beauty of line and form in a stylized depiction.

Figure 10: Detailed qualitative results: 1024px samples from LINA, Part 3.

