# OpenReview forum: "LINA: Exploring Linear Autoregressive Image Generative Models with Continuous Tokens"
_ICLR.cc/2026/Conference — Submitted to ICLR 2026_

### Official Review · Reviewer_vP1E · 2025-10-20

**Soundness:** 3
**Presentation:** 3
**Contribution:** 2
**Rating:** 6
**Confidence:** 3

**Summary:**

This submission proposes several improvements aiming to make inference of autoregressive image generative models cheaper.
To achieve this, the authors replace common attention layers in recent NOVA model with linear attention and study several design choices for that.
Namely, they compare division-based normalization with subtraction-based and add a convolutional layer to compensate the insufficient  capacity for local modeling.
In addition, learnable gates are proposed for attention heads, which seem to slightly improve the quality of generation.

The authors evaluate their model both with class-conditional generation on ImageNet and txt2img generation.

**Strengths:**

The paper considers an important problem of efficient models. The motivation for this research is clear, and the paper is mostly easy to read. Scaling behavior is examined, and it show some promise for the proposed method. Also, the comparison with other diffusion models and autoregressive models demonstrates that linear attention may be a viable alternative to the more expensive full attention.

**Weaknesses:**

1. The presented ideas are not novel per se, although the empirical study of known techniques applied to a new problem can be beneficial for the community as well.
2. It is not very clear from the text (Eq. 7), why KV gates cannot be learned just as part of the projection matrices $W_K$ and $W_V$ of the attention layer, especially if the kernel function is just a ReLU unit. While Tab. 1 demonstrates some improvement of using the gates on ImageNet, I wonder how well the model without gates performs in text-to-image task.

**Questions:**

1. While the latency of the model without optimizations is slightly greater than the latency of FlashAttn-optimized NOVA (Tab. 3), I would ask to provide the numbers for the peak memory consumption. Is my understanding correct that it should be significantly lower for the model with linear attention?
2. Kernel size of 5x5 is relatively large, even though it is a depthwise convolution. For example, this can be limiting for mobile hardware. How sensitive is the model to the design of this convolutional module?
3. Please address the question about the importance of independently learned KV gates for ReLU kernel function.

---

> ### Author Response · Authors · 2025-11-23
> **Response to Reviewer vP1E [1]**
>
> Thank you very much for your support of our work. Your recognition has given us a lot of motivation to continue improving LINA. Regardless of the final decision, we sincerely appreciate your time and encouragement. We are especially grateful that you acknowledged our **motivation**, the **clarity** of our writing, and the "**some promise**" shown by our approach.
>
> Below, we list the updates we have made based on your comments:
>
> - Appendix F (in the updated PDF): Complexity Analysis (For Q4)
> - Section 5.2 (in the updated PDF): FLOPs Comparison (For Q3)
> - Figure 6 (in the updated PDF): FLOPs comparison results. (For Q3)
>
> In the updated version of the paper, we have removed the special color formatting used in the previous draft. The **blue** highlights indicate content that was updated during the discussion stage. Next, we respond to your questions one by one.
>
>
>
> #### **Question 1: The presented ideas are not novel per se, although the empirical study of known techniques applied to a new problem can be beneficial for the community as well.**
>
> #### **Answer 1:**
>
> **Yes**, that is correct. All the components we explored in this work, including continuous-token autoregressive image generation, linear attention, division-based normalization, subtraction-based normalization, and the gate mechanism, were not originally proposed by our paper. As you mentioned, we applied these known techniques to a new problem, and we sincerely hope that this exploration can still be helpful for the community. We fully acknowledge that each of these components has been studied and examined **separately** before, but not **collectively**.
>
> We would like to explain why we believe this study is still meaningful. Our first point is it is every researcher's dream to see their works from **one problem** successfully applied to **another**. In practice, this transition is often challenging. When there is an unexplored gap between tasks, conclusions from one task may not transfer directly to another. For example, in **vision perception tasks**, *InLine* [1] adopts **subtraction-based normalization**. In contrast, by analyzing the scaling behavior, our work recommends **division-based normalization** for **autoregressive image generation**. To frankly speaking, we do not fully understand the underlying reason for this difference. It may be related to the fact that semantic confusion [1] is less severe in generative settings. Even so, we believe these observations are valuable for the development of future lightweight text-to-image architectures. Similar discussions apply to the *DWC module*, the *KV gate*, and other components. Although these techniques are not new, LINA may serve as useful **building blocks** for future research, which we believe is the core value of our contribution.
>
> Our second point is that many well-known works follow a similar spirit. They evaluate existing techniques under a new problem setting to understand their scaling behavior. For instance, *Fluid* [2] studies how discrete versus continuous tokens, and raster versus random order, affect the scaling behavior of autoregressive T2I models. The conclusions of Fluid influenced *NextStep-1* [3], a 14B-scale T2I model. Fluid also inspired our study. We wished to conduct a fair and systematic evaluation of different forms of linear attention specifically for autoregressive image generation, a setting that has not yet been carefully examined.
>
> We would like to emphasize again that we genuinely appreciate your assessment. Our intention here is only to clarify the motivation behind our work and the potential value it may offer. Thank you for your thoughtful questions, which gave us the opportunity to articulate our reasoning and our research process.
>
>
>
> [1] Bridging the Divide: Reconsidering Softmax and Linear Attention
>
> [2] Fluid: Scaling Autoregressive Text-to-image Generative Models with Continuous Tokens
>
> [3] NextStep-1: Toward Autoregressive Image Generation with Continuous Tokens at Scale

---

> ### Author Response · Authors · 2025-11-23
> **Response to Reviewer vP1E [2]**
>
> #### **Question 2: It is not very clear from the text (Eq. 7), why KV gates cannot be learned just as part of the projection matrices $W_K$ and $W_V$ of the attention layer, especially if the kernel function is just a ReLU unit. While Tab. 1 demonstrates some improvement of using the gates on ImageNet, I wonder how well the model without gates performs in text-to-image task.**
>
> #### **Answer 2:**
>
> Thank you for your thoughtful question. It is truly insightful. We address it from three perspectives:
>
> 1. Why KV gates cannot simply be learned as part of the projection matrices WKW_KWK and WVW_VWV in the attention layer.
>    The key reason is the need to model position-dependent behaviors.
> 2. The relationship between KV gates and ReLU.
> 3. The role of KV gates in text-to-image models.
>
>
>
> **Notations**
>
> In this response, we follow the notation in Section 2.2 in our main paper: assume the model uses $H$ attention heads to process $N=N_{\text{q}}+N_{\text{img}}$ tokens, where each token has a hidden dimension $D$, and the per-head dimension is $d$, satisfying $D = h d$. Denote the projection matrices for query, key, and value $Q,K, V\in\mathbb{R}^{N\times D}$ as $W_K, W_V\in\mathbb{R}^{D\times D}$. We denote the KV gates as $g^{(k)},g^{(v)}\in\mathbb{R}^{N\times 1}$, and we denote the ReLU kernel function as $\phi(\cdot)$.
>
>
>
>
>
> **1) Key difference between the KV Gate and $W_K$ and $W_V$: position dependence.**
>
> At first glance, both the KV Gate and the projection matrices $W_K$ and $W_V$ in the attention layer may look like they simply "multiply something" to their inputs. However, they **cannot be merged**.
>
> Here we describe the reason. The key and value projection matrices $W_K, W_V\in\mathbb{R}^{D\times D}$ are **position-agnostic**. Their transformation does not depend on the sequence index; every token is projected using the *same* matrix. In contrast, the KV gates $g^{(k)}, g^{(v)}\in\mathbb{R}^{N\times 1}$ are **position-specific**. Their values change with the sequence index, so they modulate each token *differently*.
>
>
>
> Here we use a *simple mathematical example* to illustrate this difference.
>
> Assume the input features are $X\in\mathbb{R}^{N\times D}$.
>
> First, $X$ is multiplied by a projection matrix $W\in\mathbb{R}^{D\times D}$, producing an intermediate result in $\mathbb{R}^{N\times D}$:
> $$
> Z=XW
> $$
> Then, this intermediate result is further modulated through an element-wise multiplication with a vector $G\in\mathbb{R}^{N\times 1}$. After this step, the final output remains in $\mathbb{R}^{N\times D}$:
> $$
> Y=Z\odot G=(XW)\odot G
> $$
> If the values $G_i$ for $i \in [1, N]$ differ across sequence positions (that is, each position in $X$ is multiplied by a different coefficient), then it is impossible to find a single projection matrix $W'\in\mathbb{R}^{D\times D}$ such that:
> $$
> XW'=(XW)\odot G
> $$
> The reason is as follows:
>
> The matrix multiplication $XW'$ applies the **same** linear transformation to **all samples (all rows)**:
> $$
> (XW')_i=X_iW'
> $$
> The scaling operation applies a **different coefficient** $G_i$ to each row, meaning that **every position** in the sequence is scaled individually.
> $$
> [(XW)\odot G]_i=G_i(X_iW)
> $$
> To simultaneously satisfy the following conditions:
> $$
> X_iW'=G_i(X_iW)
> $$
> This implies that:
> $$
> W'=G_i W
> $$
> When the coefficients $G_i$ differ across positions $i$, **there is no single matrix $W'$ that can satisfy the equivalence for all samples at the same time**.
>
>
>
> Although we cannot merge them into a single matrix, we can still write the operation as a whole:
> $$
> Y = \mathrm{diag}(G) X W
> $$
> Here, $\mathrm{diag}(G)\in \mathbb{R}^{N\times N}$ is the row-scaling matrix.
>
> However, even with this formulation, we still cannot merge $G$ and $W$ unless all $G_i$ take the same value across positions. In practice, this condition is too restrictive. As supporting evidence, we provide detailed visualizations of the KV gates in **Figure 4-(b)** and **Figure 7** of the main paper.

---

> ### Author Response · Authors · 2025-11-23
> **Response to Reviewer vP1E [3]**
>
> **2) Relationship between the KV gate and ReLU**
>
> The KV gate works by applying a function to the key and value, which then modulates the linear attention terms $M$ and $z$:
>
> $$
> {\tilde{K}_j}={g_j^{(k)}}\phi(K_j), \qquad
> {\tilde{V}_j}={g_j^{(v)}}V_j,\qquad \textit{for}\qquad j\in \left[1,N\right]
> $$
>
>
>
> $$
> M=\sum_ {j=1}^{N}{
> {\tilde{K}_ j}^\top{\tilde{V}_ j}
> }=
> \sum_ {j=1}^{N}{{g_ j^{(k)}}{g_ j^{(v)}}M_ j},\qquad
> z^{(\texttt{d})}=\sum_ {m=1}^{N}{{\tilde{K}_ m}^\top},\qquad
> O^{(\texttt{d})}_ i
> = \frac{\phi(Q_ i) M}{\phi(Q_ i) z}.
> $$
>
>
>
> Here, we view $M^{(\texttt{d})}$ as the *memory* term of linear attention, while $z^{(\texttt{d})}$ contributes to the normalization term $\phi(Q_i)\, z^{(\texttt{d})}$. In vanilla linear attention, the memory is simply summed over all positions. Our motivation for introducing the KV gate is to distinguish the memory at different positions and enable more flexible modulation.
>
>
>
> The role of the ReLU function is different: it transforms the query and key through $\text{Sim}(Q,K) = \phi(Q)\phi(K)^\top$
> to approximate the similarity measure of softmax attention, $\text{Sim}(Q,K) = \exp(QK^\top/\sqrt{d})$.
>
>
>
> However, using the ReLU function alone cannot flexibly modulate the memory across positions. Our KV gate is designed specifically to fill this gap, and it can be used together with ReLU.
>
>
>
>
>
> **3) The role of the KV gate in text-to-image models**
>
> Regarding the impact of KV gates on text-to-image generation, we acknowledge that our computational budget was very limited. :( As a result, we were only able to conduct ablation studies on ImageNet. Nevertheless, for the KV gates used in text-to-image models, we provide detailed visualization analyses in Figure 4-(b) and Figure 7 of the main paper.
>
> We summarize several interesting observations below:
>
> 1. Different layers learn different KV gate patterns. For example, in layer 10, head 1 shows relatively stable KV gate values across positions, while in layer 16, head 1 the values fluctuate much more strongly.
> 2. Different heads in the same layer may serve different roles. In layer 16, head 1 shows large fluctuations, whereas head 6 and head 11 vary much more smoothly across positions.
> 3. In terms of value ranges, many KV gates fall within (−1, 1), such as layer 10 head 6, layer 4 head 1, and layer 4 head 11. This may suggest that, in some cases, the model applies token-wise attenuation when computing $M^{(\texttt{d})}$ and $z^{(\texttt{d})}$.
>
> These observations indicate that KV gates indeed **learn meaningful patterns and behaviors**.
>
> We hope that these visualization results offer an informative perspective and provide supportive evidence for understanding the role of the KV gate.

---

> ### Author Response · Authors · 2025-11-23
> **Response to Reviewer vP1E [4]**
>
> #### **Question 3: While the latency of the model without optimizations is slightly greater than the latency of FlashAttn-optimized NOVA (Tab. 3), I would ask to provide the numbers for the peak memory consumption.**
>
> #### **Answer 3:**
>
> Thank you very much for this constructive reminder. As a paper that focuses on lightweight architectures, we fully agree that, beyond latency, metrics such as FLOPs and GPU memory usage are indeed important for evaluating efficiency.
> **We have completed the FLOPs and GPU memory measurements and provide the results below.**
>
> We have also updated the FLOPs-related content in **Section 5.2** accordingly. If you have any further questions or concerns, please do not hesitate to let us know. We sincerely appreciate your suggestions, as they have helped us improve our work greatly. Thank you again!
>
>
>
> ##### **FLOPs Comparison**
>
> We compare the FLOPs of linear attention and full attention. We report the FLOPs of **a single** attention module under the following configuration: batch size of 1, sequence length of 5120, hidden dimension of 1536, and 16 attention heads. This setup matches how LINA-H operates at a resolution of 1024px.
>
> The linear attention module we evaluate adopts *division-based normalization* in the LINA *Decoder* and integrates both the *DWC module* and the *KV gate* proposed in this work. FLOPs are measured using the fvcore [1] library.
>
> The results are presented in **Table 1** below. A full-attention module requires approximately ~129 GFLOPs, whereas the linear-attention module requires only ~50 GFLOPs. This corresponds to a reduction of about **~61%** in computation, highlighting the efficiency benefits of our LINA. Importantly, despite this substantial reduction in FLOPs, the text-to-image performance of LINA remains competitive with the full-attention-based NOVA.
>
> For clarity, we note that the *DWC module* and the *KV gate* are used only in the LINA *Decoder*, and are not applied in the *Encoder* or the *Connector*.
>
> [1] https://github.com/facebookresearch/fvcore
>
>
>
> **Table 1: FLOPs comparison results. Compared with the full attention, a single linear attention module reduces FLOPs by about ~61%, showing computation efficiency.**
>
> | Module                                     | GFLOPs |
> | ------------------------------------------ | ------ |
> | Linear Attention (w/ DWC module & KV gate) | 49.99  |
> | Full Attention                             | 128.85 |
>
>
>
> ##### **GPU Memory Comparison**
>
> We compared the GPU usage when generating a 1024px image using NOVA with **Flash Attention** and our LINA model with **linear attention**. The results are shown in **Table 2**, where we report both allocated and reserved memory. As the table shows, without applying any extra optimization techniques, our LINA model achieves GPU memory usage that is comparable to the Flash Attention version of full attention.
>
> We fully acknowledge that, in terms of GPU memory, LINA does not show a clear advantage over Flash Attention. However, we would like to gently share our perspective:
>
> Flash Attention is a **highly optimized** and mature implementation designed specifically for **full attention**, while our linear attention is implemented in plain PyTorch *without any additional acceleration*. Given this gap in engineering maturity, it is not surprising that their memory usage appears similar. In fact, this result highlights an important point: **linear attention already matches Flash Attention *before* any dedicated optimization**, showing its inherent algorithmic efficiency. Demonstrating this **algorithm-level potential** is one of the *key contributions* of our work.
>
> We believe that developing specialized acceleration techniques for linear attention is an exciting direction for future research, and we hope our findings can motivate further progress in this area.
>
>
>
>
>
>
>
> **Table 2: GPU memory comparison results.**
>
> | Model                      | Allocated  | Reserved   |
> | -------------------------- | ---------- | ---------- |
> | LINA (w/ Linear Attention) | 5406.55 MB | 8072.00 MB |
> | NOVA (w/ Flash Attention)  | 5401.96 MB | 7552.00 MB |

---

> ### Author Response · Authors · 2025-11-23
> **Response to Reviewer vP1E [5]**
>
> #### **Question 4: Kernel size of 5x5 is relatively large, even though it is a depthwise convolution. For example, this can be limiting for mobile hardware. How sensitive is the model to the design of this convolutional module?**
>
> #### **Answer 4:**
>
>
>
> ##### **1) Is 5×5 DWC module computation-intensive? No. The FLOPs of convolution are significantly lower than those of linear attention.**
>
> We include a detailed **complexity analysis** of both modules here, and we have also updated **Appendix F** of the paper accordingly.
>
> ##### **Complexity Analysis**
>
> We follow the notation in Section 2.2 in our main paper: assume the model uses $H$ attention heads to process $N=N_{\text{q}}+N_{\text{img}}$ tokens, where each token has a hidden dimension $D$, and the per-head dimension is $d$, satisfying $D = h d$.
>
> **Linear Attention**
>
> The computation of the per-token output $O^{(\texttt{d})}_i$ for linear attention with division-based normalization can be expressed as:
>
>
> $$
> O^{(\texttt{d})}_ i
> = \frac{\phi(Q_i)\left(\sum_ {j=1}^{N} \phi(K_j)^\top V_j \right)}
>        {\phi(Q_i)\left(\sum_ {m=1}^{N} \phi(K_m)^\top \right)}.
> $$
>
>
> Theoretically, for the numerator, we have:
> $$
> \underbrace{\phi(Q_i)}_ {\mathbb{R}^{1\times d}}
> \left(
> \sum_{j=1}^{N}
>     {
>     \underbrace{\phi(K_j)^\top}_ {\mathbb{R}^{d\times 1}}
>     \underbrace{V_j}_ {\mathbb{R}^{1\times d}}
>     }
> \right)
> \longrightarrow
> \mathbb{R}^{1\times d}.
> $$
>
>
> where, first, a $d \times 1$ matrix is multiplied by a $1 \times d$ matrix, which costs $\mathcal{O}(d^{2})$.
> Then, a $1 \times d$ matrix is multiplied by an a $d \times d$ matrix, which also costs $\mathcal{O}(d^{2})$.
>
> For the denominator, we have:
> $$
> \underbrace{\phi(Q_i)}_ {\mathbb{R}^{1\times d}}
> \left(
> \sum_{m=1}^{N}
>     {
>     \underbrace{\phi(K_m)^\top}_ {\mathbb{R}^{d\times 1}}
>     }
> \right)
> \longrightarrow
> \mathbb{R}^{1\times 1}.
> $$
>
>
>
>
> where, a $1 \times d$ matrix is multiplied by a $d \times 1$ matrix, which costs $\mathcal{O}(d)$, which is negligible in the big-$\mathcal{O}$ sense.
>
> Computing the final output $O^{(\texttt{d})}_ i$ also costs $\mathcal{O}(d)$, which is negligible in the big-$\mathcal{O}$ sense.
>
> Thus, the per-token complexity is at most $\mathcal{C}_ {attn, token}=2d^{2}=\mathcal{O}(d^{2})$.
>
> For a single attention head processing $N$ tokens, the total cost is at most:
>
>
> $$
> \text{Linear attention, per-head:}\qquad
> \mathcal{C}_ {attn, head}
> = N \mathcal{C}_ {attn, token}
> = \mathcal{O}(N d^{2}).
> $$
>
>
>
>
> For the whole linear attention with $h$ heads (and hidden dimension $D = h d$), the total cost is:
> $$
> \text{Linear attention:}\qquad
> \mathcal{C}_{attn}
> = h \mathcal{C}_{attn, head}
> = \mathcal{O}(N D d).
> $$
>
>
>
>
> ##### **Depthwise Convolution**
>
> The DWC module applies a $k \times k$ depthwise convolution to $N_{\text{img}}$ image tokens only, with a cost of:
> $$
> \text{DWC module:}\qquad
> \mathcal{C}_ {dwc}
> = \mathcal{O}(N_ {\text{img}} D k^2).
> $$
>
>
> The ratio between the computational cost of the **DWC module** and that of **linear attention** is:
> $$
> \frac{\mathcal{O}(N_ {\text{img}} D k^2)}{\mathcal{O}(N D d)}
> \leq
> \frac{k^2}{d}.
> $$
>
>
> Note that the DWC module processes only $N_{\text{img}}$ image tokens, while the linear attention processes both $N_{\text{q}}$ query tokens and $N_{\text{img}}$ image tokens ($N=N_{\text{q}}+N_{\text{img}}$).
>
> For our text-to-image model LINA-H, we have $D = 1536$ and $h = 16$, so the per-head dimension is $d = 96$.
>
>
>
> When using a kernel size of $5$, we have **$k^{2}/d = 0.26$**. When using a kernel size of $3$, we have **$k^{2}/d = 0.09$**. Regardless of whether the kernel size is $5$ or $3$, the **additional FLOPs** it introduces are still **significantly lower** than those of linear attention.

---

> ### Author Response · Authors · 2025-11-23
> **Response to Reviewer vP1E [6]**
>
> ##### **2) Why we use 5×5 here?**
>
> Our intuition is that **high-resolution image generation tasks naturally produce much larger feature maps, which may benefit from slightly larger convolution kernels**.
>
> For example, in a standard ViT with an input resolution of 224 and a patch size of 16, the resulting feature map has spatial dimensions of **14×14**. In this setting, a **3×3** convolution kernel is generally sufficient. Indeed, existing models designed for visual perception, such as *Agent Attention* [1], also adopt 3×3 convolutions.
>
> However, in high-resolution image generation (for instance, LINA-H operating at 1024px), the input to the DWC module has spatial dimensions of **64×64**. In this scenario, we hypothesize that slightly increasing the kernel size (e.g., from **3×3** to **5×5**) can help the autoregressive model better utilize nearby known tokens when predicting the next token group. We provide a more detailed discussion in Section 3.2.2. Consistent with this intuition, some image generation models such as *LiT* [2] indeed employ 5×5 convolution kernels.
>
> In summary, the additional FLOPs introduced by moderately larger kernels remain **far smaller** than the FLOPs of linear attention. From a task-driven perspective, high-resolution image generation produces large feature maps, which may benefit from using larger convolution kernels (e.g., 5×5).
>
>
>
> [1] Agent Attention: On the Integration of Softmax and Linear Attention
>
> [2] LiT: Delving into a Simple Linear Diffusion Transformer for Image Generation

---

### Official Review · Reviewer_wwwH · 2025-11-01

**Soundness:** 3
**Presentation:** 3
**Contribution:** 2
**Rating:** 4
**Confidence:** 4

**Summary:**

This paper investigates how to design efficient linear attention mechanisms specifically for autoregressive image generative models with continuous tokens, a setting that differs from ViT-based perception or diffusion-based generation due to its step-wise causal decoding process. The authors propose LINA, a pure linear-attention-based autoregressive generator capable of producing high-resolution (1024×1024) images with competitive quality and faster sampling compared to full-attention counterparts.

**Strengths:**

* This paper provides a feasibility study of applying linear attention to autoregressive image generative models, decomposing the problem into several sub-components and addressing them step by step.
* The authors propose **LINA**, a text-to-image model that aims to balance efficiency and fidelity under a pure linear attention architecture.
* Experimental coverage is extensive, and the design choices appear to be systematically ablated and reproducible.

**Weaknesses:**

1. **Limited novelty.**
   The main contributions primarily consist of empirical combinations and refinements of existing techniques:
   i) exploring normalization strategies for linear attention and introducing depthwise convolution to compensate for locality;
   ii) introducing a KV gate module.
   While the observation that linear attention lacks local inductive bias and thus benefits from depthwise convolution is somewhat insightful, the work still lacks a theoretically grounded innovation or a fundamentally new mechanism.

2. **Presentation issues leading to a technical report style rather than a focused scientific argument.**

   * The preliminary section is overly long and the related work is placed at the end, which makes the structure feel misaligned. Foundational content such as MAR models does not require such an extended exposition.
   * The two normalization variants, claimed as core design decisions, could be introduced more concisely under the Methods section without heavy narrative buildup.
   * Motivation is repeated across Introduction and Methods, leading to redundancy.
   * The paper jumps between methodology, implementation details, and experimental settings, instead of clearly separating method description and empirical validation.

3. **Experimental presentation could be improved.**
   The best results in tables should be bolded, and if possible, second-best results could be underlined to enhance readability.

4. **Some generation results lack semantic consistency.**
   For example, in Figure 7 of the appendix, the first example fails to preserve the concept "jetpack", which suggests that semantic grounding is not always stable.

**Questions:**

Please refer to the "Weakness".

---

> ### Author Response · Authors · 2025-11-23
> **Response to Reviewer wwwH [1]**
>
> Thank you very much for recognizing the effort we put into our experiments and describing them as “extensive.” We are also sincerely grateful for your kind assessment of our design choices as “systematically ablated.” Your comments are highly professional, especially the many suggestions you provided regarding the presentation of the paper.
>
> We are truly glad that our work received such valuable writing-related feedback, and we have carefully revised the manuscript according to your suggestions. We have now submitted the updated version.
>
> We also noticed that most of your suggestions focus on writing and presentation, and that you raised relatively few concerns about the experimental results themselves. We deeply appreciate your constructive feedback and fully agree with your thoughtful critiques.
>
> In the updated version of the paper, we have removed the special color formatting used in the previous draft. The **blue** highlights indicate content that was updated during the discussion stage. Next, we respond to your questions one by one.
>
>
>
> #### **Question 1.1: Limited novelty. The main contributions primarily consist of empirical combinations and refinements of existing techniques: i) exploring normalization strategies for linear attention and introducing depthwise convolution to compensate for locality; ii) introducing a KV gate module. While the observation that linear attention lacks local inductive bias and thus benefits from depthwise convolution is somewhat insightful, the work still lacks a theoretically grounded innovation or a fundamentally new mechanism.**
>
> #### **Answer 1.1:**
>
> We sincerely appreciate your comments. They are highly professional and very fair. You are absolutely right that all the components we experimented with in this work, including continuous-token autoregressive image generation, linear attention, division-based normalization, subtraction-based normalization, and the gate mechanism, are **NOT** originally from this paper. As you noted, our contribution lies in the empirical combinations and refinements of these existing techniques. Indeed, each component has been explored in prior work **separately**, but not **collectively**.
>
> Why do we believe this still matters? We would like to share three humble thoughts.
>
> **First**, it is every researcher’s dream to see methods from one domain successfully adapted to a new task. Yet this process is rarely smooth, because *conclusions from Task A often do not transfer directly to Task B when there exists an unstudied gap*. For example, **InLine** [1] adopts subtraction-based normalization for visual perception. In contrast, our scaling analysis suggests that division-based normalization is more suitable for autoregressive image generation. We are not fully sure why, perhaps because the "semantic confusion" issue [1] is less severe in image generation. Even so, these findings may guide future lightweight T2I architectures. Similar reflections apply to the DWC module and the KV gate. While none of these techniques are new in isolation, we hope LINA can serve as building blocks for future work, which we believe is its true value.
>
> **Second**, many influential papers take a **similar approach**: evaluating the scaling behavior of existing techniques under a new problem. For instance, **Fluid** [2] studies how discrete versus continuous tokens, and raster versus random ordering, affect scaling behavior in autoregressive T2I modeling. Its conclusions also inspired the large-scale **NextStep-1** [3]. Fluid similarly inspired our study, motivating us to conduct a fair comparison of different linear-attention variants specifically under autoregressive image generation.
>
> **Third**, regarding theoretical innovation, we fully acknowledge that our work is limited in this respect.
>
> Our main contribution is that we are, to the best of our knowledge, the **first** to adapt well-established ideas from visual perception and diffusion models into the new setting of autoregressive image generation, and to demonstrate a practical and successful implementation. We hope this effort can serve as a useful foundation for future research. We hope that our work can serve as a ready-to-use paradigm that supports future theoretical exploration. In this sense, its value lies in **adapting concepts to a new task and providing a solution that researchers can safely build upon**.
>
> We want to emphasize, with respect and sincerity, that we genuinely appreciate your evaluation. Our goal here is simply to clarify the intention behind this work and what we believe it can contribute. Thank you very much for your question, which gave us the opportunity to express our thoughts and experience.
>
>
>
> [1] Bridging the Divide: Reconsidering Softmax and Linear Attention
>
> [2] Fluid: Scaling Autoregressive Text-to-image Generative Models with Continuous Tokens
>
> [3] NextStep-1: Toward Autoregressive Image Generation with Continuous Tokens at Scale

---

> ### Author Response · Authors · 2025-11-23
> **Response to Reviewer wwwH [2]**
>
> #### **Question 2.1: Presentation issues. The preliminary section is overly long and the related work is placed at the end, which makes the structure feel misaligned. Foundational content such as MAR models does not require such an extended exposition.**
>
> #### **Answer 2.1:**
>
> Thank you for your suggestion.
>
> **We have moved the related work section to follow directly after the introduction**.
>
> **We have also simplified the MAR-related content in the preliminary section**.
>
> We sincerely appreciate your feedback, and we have submitted the updated version of the paper.
>
>
>
>
>
> #### Question 2.2: Presentation issues. The two normalization variants, claimed as core design decisions, could be introduced more concisely under the Methods section without heavy narrative buildup.
>
> #### Answer 2.2:
>
> Thank you very much for your suggestion. Since both division-based normalization and subtraction-based normalization were not introduced for the first time in our work, we placed them in the preliminary section. We used roughly forty lines to cover:
>
> - the notation used in our paper
> - the formulaic implementations of both methods
>
> We tried our best to use the minimum necessary space while still giving a complete and clear description.
>
> If you feel that any part is still redundant, we would be truly grateful if you could point it out directly. Your feedback will help us further improve the clarity of our manuscript.
>
>
>
>
>
>
>
> #### Question 2.3: Presentation issues. Motivation is repeated across Introduction and Methods, leading to redundancy.
>
> #### Answer 2.3:
>
> Thank you very much for your suggestion.
>
> We would like to gently clarify that the motivation paragraph in the method section highlights three key research questions in our work:
>
> - linear attention paradigm choice
> - locality choice
> - memory management
>
> These points are indeed mentioned in the introduction. We restated them in a clear and explicit way in the method section to help readers who may start reading directly from this part, and to create a clear connection with the introduction. In fact, many papers adopt a similar practice, such as [1].
>
> **We have now simplified the motivation paragraph in the method section**. We sincerely appreciate your careful feedback and guidance.
>
>
>
> [1] Agent Attention: On the Integration of Softmax and Linear Attention
>
>
>
>
>
> #### Question 2.4: Presentation issues. The paper jumps between methodology, implementation details, and experimental settings, instead of clearly separating method description and empirical validation.
>
> #### Answer 2.4:
>
> Thank you very much for your thoughtful response.
>
> In fact, presenting ablation details and findings before the main experiments is a **common structure** used in many exploratory papers. This applies to works in **vision perception models [1,2,4]**, **image generation models [4]**, and **LLM research [5]**. Our paper follows this writing pattern as well.
>
> At the same time, we fully agree with your perspective that clearly separating the method section from the experiment section is also an excellent choice. In our case, the central question we aim to explore is how to effectively adapt existing techniques such as linear attention and the DWC module to the new task of autoregressive image generation. Therefore, we placed this roadmap before the main experiments to help readers understand the reasoning behind our design choices.
>
> We truly hope this explanation addresses your concern. If anything in our response remains unclear, we would be very grateful for your further comments.
>
>
>
> [1] RIFormer: Keep Your Vision Backbone Effective While Removing Token Mixer
>
> [2] A ConvNet for the 2020s
>
> [3] LiT: Delving into a Simple Linear Diffusion Transformer for Image Generation
>
> [4] Scaling Up Your Kernels to 31x31: Revisiting Large Kernel Design in CNNs
>
> [5] MobileLLM: Optimizing Sub-billion Parameter Language Models for On-Device Use Cases

---

> ### Author Response · Authors · 2025-11-23
> **Response to Reviewer wwwH [3]**
>
> #### **Question 3: Experimental presentation could be improved. The best results in tables should be bolded, and if possible, second-best results could be underlined to enhance readability.**
>
> #### **Answer 3:**
>
> Thank you very much for your reminder. We have updated the class-conditional ImageNet benchmark results in Table 2, as well as the text-to-image results in Table 4. Following your suggestion, we highlighted the best results in bold and marked the second-best results with underlines to improve readability.
>
> The updated PDF has been submitted.
>
> Thank you again for helping us improve the presentation of our work. If you have any further questions or suggestions, we would be grateful to hear them.

---

### Official Review · Reviewer_7uD7 · 2025-11-01

**Soundness:** 3
**Presentation:** 4
**Contribution:** 3
**Rating:** 4
**Confidence:** 3

**Summary:**

This paper proposes LINA, a linear autoregressive image generation architecture employing continuous tokens, designed for efficiency and competitive image quality at resolutions up to 1024×1024. The authors comprehensively investigate design paradigms for linear attention—specifically, division-based vs. subtraction-based normalization—and the impact of locality augmentation using depthwise convolution (DWC). They introduce a data-independent KV gate mechanism to flexibly manage memory within linear attention layers. Extensive ablations, scaling experiments, and qualitative/quantitative evaluations are conducted on class-conditional and text-to-image generation benchmarks, with comparisons against recent diffusion and autoregressive baselines.

**Strengths:**

1. The empirical study rigorously dissects the scaling behavior of different linear attention paradigms, providing actionable guidance on architecture design for autoregressive image generation.
2. The paper is well written, with rigorous structure and clear logical flow.
3. The experiments are comprehensive and carefully conducted, providing solid empirical support for the proposed approach.

**Weaknesses:**

1. Limited Novelty in Core Mechanisms: While the empirical study on scaling and the introduction of KV gate are well-executed, the architectural changes (division-based normalization, DWC, gating) largely package and extend existing ideas from linear attention literature (Katharopoulos et al., Han et al., Qiu et al., Yang et al.). There is limited original theoretical derivation or clear demonstration that the KV gate delivers fundamentally different benefits compared to gating or locality modules already studied elsewhere.
2. Despite referencing recent diffusion and linear attention efforts (Section 5), the paper insufficiently contextualizes and empirically compares to several directly related works on continuous-token autoregressive image generation.
3.  Table 3, comparing LINA’s latency to FlashAttention, reveals little actual speedup—22s vs. 20s at 1024px, with FlashAttention on the full-attention baseline—and the observed latency parity is not explained in terms of FLOPs or memory.
4. Minor Issues: Occasional dense or overloaded notation in equations (e.g., Section 3.3), some grammatical/typo issues persist (e.g., “Addition-based” sometimes called “subtraction-based”), reference style inconsistent in places, and a few claims (e.g., “negligible parameter overhead”) are asserted without accompanying numbers or clear resource comparison.

**Questions:**

Please provide responses to the issues raised in my Weaknesses section.

---

> ### Author Response · Authors · 2025-11-23
> **Response to Reviewer 7uD7 [1]**
>
> We sincerely thank you for recognizing our work and for offering such encouraging comments, including “with rigorous structure and clear logical flow” and “solid empirical support.” Your positive feedback truly means a lot to us.
>
> At the same time, we feel that the questions you raised are both highly professional and very insightful. When we first read them, we immediately realized that these were issues we needed to consider with great seriousness.
>
> In response, we have made sincere and substantial efforts to address your concerns as carefully as we can. We genuinely hope that our revisions help resolve the points you highlighted.
>
> Below, we list the updates we have made based on your comments:
>
> - Appendix F (in the updated PDF): Complexity Analysis (For Q3)
> - Section 5.2 (in the updated PDF): FLOPs Comparison (For Q3)
> - Table 4 (in the updated PDF): Comparison of GenEval results. (For Q2)
> - Figure 6 (in the updated PDF): FLOPs comparison results. (For Q3)
> - Table 10 (in the updated PDF): Training setting of our text-to-image generation LINA. (For Q2)
>
> In the updated version of the paper, we have removed the special color formatting used in the previous draft. The **blue** highlights indicate content that was updated during the discussion stage. Next, we respond to your questions one by one.

---

> ### Author Response · Authors · 2025-11-23
> **Response to Reviewer 7uD7 [2]**
>
> #### **Question 1.1: Limited Novelty in Core Mechanisms: While the empirical study on scaling and the introduction of KV gate are well-executed, the architectural changes (division-based normalization, DWC, gating) largely package and extend existing ideas from linear attention literature (Katharopoulos et al., Han et al., Qiu et al., Yang et al.).**
>
> #### **Answer 1.1:**
>
> Thank you very much for your professional and thoughtful comments. We sincerely appreciate your suggestions.
>
> It is true that all components used in our work — continuous-token autoregressive image generation, division-based normalization, DWC, and gating — are **NOT** originally proposed in this paper. As you pointed out, we build upon several existing ideas explored in prior works [1,2,3]. We openly acknowledge that each of these components has been studied separately, but not collectively.
>
> What, then, is the value of putting them together? We would like to humbly share two points of our perspective:
>
> **First**, having a method developed in one domain successfully apply to a different task is something every researcher hopes for. However, in practice, this transition is rarely straightforward. The main reason is simple: when there exists an *under-explored gap* between two tasks, conclusions drawn from Task A cannot always be directly transferred to Task B.
>
> For example, consider linear attention. In vision perception tasks, *InLine* [4] adopts *subtraction-based* normalization. In contrast, through analyzing the scaling behavior, our work recommends using *division-based* normalization for autoregressive image generation. To be completely honest, we do not yet fully understand **why** this happens. One possibility is that the "semantic confusion" problem discussed in [4] is less severe in image generation. But regardless of the exact underlying reason, we believe these findings are **meaningful** for future research on **efficient text-to-image architectures**.
>
> A similar situation appears with the DWC module and the KV gate. Although these components are not novel by themselves, our study shows how they behave *together* in this specific and challenging setting. We humbly believe that LINA can serve as a set of useful **building blocks** for future work. In our view, this is where the true value of our **contribution** lies.
>
> **Second**, many influential works in our community take a *similar* approach. They focus on empirically evaluating how existing techniques behave when scaled to a new problem setting. For example, *Fluid* [5] carefully examined how different design choices, such as **discrete** versus **continuous tokens** or **raster** versus **random ordering**, affect the scaling behavior of autoregressive text-to-image models. The conclusions drawn in Fluid later influenced *NextStep-1* [6], a large 14B T2I model.
>
> Fluid’s way of thinking also inspired our work. Autoregressive image generation is still a relatively new problem, so we genuinely wanted to understand which type of **linear attention** is most appropriate and to compare them in a *fair and consistent manner*. Our goal was to offer clarity for future research on this topic.
>
> Thank you very much for your question. It gives us a valuable opportunity to share our thoughts and the reasoning behind our work with sincerity.
>
>
>
> [1] Transformers are RNNs: Fast Autoregressive Transformers with Linear Attention
>
> [2] FLatten Transformer: Vision Transformer using Focused Linear Attention
>
> [3] Gated Linear Attention Transformers with Hardware-Efficient Training
>
> [4] Bridging the Divide: Reconsidering Softmax and Linear Attention
>
> [5] Fluid: Scaling Autoregressive Text-to-image Generative Models with Continuous Tokens
>
> [6] NextStep-1: Toward Autoregressive Image Generation with Continuous Tokens at Scale

---

> > ### Author Response · Authors · 2025-11-23
> > **Response to Reviewer 7uD7 [8]**
> >
> > #### **Question 4: Minor Issues: Occasional dense or overloaded notation in equations (e.g., Section 3.3), some grammatical/typo issues persist (e.g., “Addition-based” sometimes called “subtraction-based”), reference style inconsistent in places, and a few claims (e.g., “negligible parameter overhead”) are asserted without accompanying numbers or clear resource comparison.**
> >
> > #### **Answer 4:**
> >
> > Thank you very much for the helpful reminder.
> >
> > For the notation issue, we now consistently use the term "subtraction-based".
> >
> > In **Lines 262–264** (in the updated PDF), we added concrete numbers (the model size is approximately 0.4B parameters, while the DWC module introduces only 0.31M) to support the claim of “negligible parameter overhead.”
> >
> > In **Lines 355–357** (in the updated PDF), we also included the corresponding numbers (the model size is approximately 0.4B parameters, while the KV gate introduces only 0.12M) to justify the "negligible parameter overhead".
> >
> > We updated some of the notations in the main text formulas.
> >
> > These issues have been corrected in the updated version of the paper. We sincerely appreciate your valuable suggestions.

---

> ### Author Response · Authors · 2025-11-23
> **Response to Reviewer 7uD7 [3]**
>
> #### **Question 1.2: There is limited original theoretical derivation or clear demonstration that the KV gate delivers fundamentally different benefits compared to gating or locality modules already studied elsewhere.**
>
> #### **Answer 1.2:**
>
> Thank you very much for your question. It is a timely point that gives us the opportunity to clarify the relationship between the **KV gate** in our work and the **gated linear attention** mechanisms explored in prior studies.
>
> The main contribution of our KV gate is that **we provide a practical and effective way to regularize the memory through gating within bidirectional linear attention** for autoregressive image generation.
>
> We denote the KV gates as $g^{(k)}, g^{(v)} \in \mathbb{R}^{N \times 1}$, and the memory for division-based normalization linear attention as $M$.
>
>
>
> **Gated Linear Attention Suits Causal Mode Attention**
>
> We would like to openly acknowledge that gated linear attention, as an alternative to full attention, has already been widely validated in LLMs [1,2]. In LLMs, the commonly used form is **causal linear attention**, where the memory can be directly controlled through a gate $\alpha_t \in \mathbb{R}$:
> $$
> M_t=\alpha_tM_{t-1}+K_t^\top{V_t} =\sum_{j=1}^{t}{
> (\prod_{s=j+1}^t \alpha_s){K_j}^\top{V_j}
> }.
> $$
>
> In causal linear attention, the memory contribution for the $i$-th token is weighted by a cumulative product term $\prod_{s=j+1}^t \alpha_s$. This design choice may be closely related to the characteristics of language modeling tasks.
>
>
>
> **KV Gate Suits Bidirectional Mode Attention**
>
> However, in image generation, LINA uses **bidirectional linear attention**. As a result, LINA does not rely on the cumulative product term $\prod_{s=j+1}^t \alpha_s$ when computing the memory. Instead, each token’s memory is scaled by its own individual coefficient.
>
> To achieve this in practice, we introduce the KV gate, which applies token-specific operations to the key and value representations, enabling effective control over the memory:
> $$
> {\tilde{K}_j}={g_j^{(k)}}\phi(K_j), \qquad
> {\tilde{V}_j}={g_j^{(v)}}V_j,\qquad \textit{for}\qquad j\in \left[1,N\right]
> $$
>
> $$
> M=\sum_ {j=1}^{N}{
> {\tilde{K}_ j}^\top{\tilde{V}_ j}
> }=\sum_ {j=1}^{N}{{g_j^{(k)}}{g_j^{(v)}}M_j}.
> $$
>
> As a result, our KV gate can be viewed as **a successful adaptation and practical realization of gated linear attention in the bidirectional linear attention setting.** We believe this adaptation is the main contribution of the KV gate. In the future, KV gates may be useful in scenarios where bidirectional linear attention is required and flexible control of the linear-attention memory is important. Such scenarios naturally appear in many image generation tasks.
>
> We sincerely appreciate your question, which gave us the opportunity to further clarify the role of the KV gate.
>
>
>
>
>
>
> [1] Gated Linear Attention Transformers with Hardware-Efficient Training
>
> [2] Gated Delta Networks: Improving Mamba2 with Delta Rule

---

> ### Author Response · Authors · 2025-11-23
> **Response to Reviewer 7uD7 [4]**
>
> #### **Question 2: Despite referencing recent diffusion and linear attention efforts (Section 5), the paper insufficiently contextualizes and empirically compares to several directly related works on continuous-token autoregressive image generation.**
>
> #### **Answer 2:**
>
> Thank you very much for your question. In the updated version of our paper, we have:
>
> 1. compared our method with several works discussed in Section 5
> 2. updated the results of our LINA-H text-to-image model
>
> We explain these points in more detail below.
>
>
>
> ##### **1) The Works Compared in Section 5**
>
> The works we list in Section 5 are as follows.
>
> [1] Autoregressive Image Generation without Vector Quantization
>
> [2] Fluid: Scaling Autoregressive Text-to-image Generative Models with Continuous Tokens
>
> [3] NextStep-1: Toward Autoregressive Image Generation with Continuous Tokens at Scale
>
> [4] Autoregressive Video Generation without Vector Quantization
>
> [5] Transformers are RNNs: Fast Autoregressive Transformers with Linear Attention
>
> [6] Rethinking Attention with Performers
>
> [7] Gated Linear Attention Transformers with Hardware-Efficient Training
>
> [8] EfficientViT: Multi-Scale Linear Attention for High-Resolution Dense Prediction
>
> [9] FLatten Transformer: Vision Transformer using Focused Linear Attention
>
> [10] SANA: Efficient High-Resolution Image Synthesis with Linear Diffusion Transformers
>
> [11] LiT: Delving into a Simple Linear Diffusion Transformer for Image Generation
>
> [12] DiG: Scalable and Efficient Diffusion Models with Gated Linear Attention
>
> [13] Efficient Diffusion Transformer with Step-wise Dynamic Attention Mediators
>
>
>
>
>
> The methods **[2, 4, 10]** have already been compared in **Table. 4** of the updated manuscript (text-to-image generation results).
>
> The methods **[1, 11]** have been included in **Table. 2** of the updated manuscript (class-conditional ImageNet generation results).
>
> We would like to clarify that references **[5, 6, 7]** are **not** designed for autoregressive image generation. As a result, their results cannot be directly compared to the autoregressive scenario we study.
>
> We would like to note that references **[8, 9]** are designed for vision perception tasks. Their goals and settings differ from those of autoregressive image generation. As a result, their results cannot be directly compared to the autoregressive scenario we study.
>
> The results in **[12, 13]** have already been updated and included in **Table. 2** of the revised manuscript.
>
> Reference **[3]** is a large model with around **14B** parameters, which is far larger than our **1.5B** LINA-H model. For this reason, we did not include it in our comparisons.

---

> ### Author Response · Authors · 2025-11-23
> **Response to Reviewer 7uD7 [5]**
>
> ##### **2) The Updated Results of LINA-H Text-to-image Model**
>
> After submitting the draft version, we noticed a potential issue that may affect the performance of the 1024px LINA-H model: **overfitting**. In the draft version, we mistakenly trained the 512→1024px stage for **700k** iterations, which unintentionally led to **serious overfitting**. We did not notice this issue when preparing the draft.
>
> During the rebuttal period, we updated the LINA-H model. In the new version, we observed that training the 512→1024px stage for **only 50k** or **60k** iterations already brings a significant performance gain (**GenEval 0.72 in the updated version vs. 0.67 in the draft**, at 1024px), as shown in **Table 1**.
> All other settings, including the learning rate and architecture, remained unchanged.
>
> In addition, the 1.4B LINA model achieves a GenEval score of **0.74** at 512px, outperforming the 10.5B Fluid model and matching the full-attention NOVA model at the same resolution (**0.74 vs. 0.75**, at 512px).
> We have incorporated all updated results into Table. 4 of the updated manuscript.
>
> These results improve the competitiveness of our LINA model on the evaluation metrics, making LINA — a fully linear-attention-based and efficient text-to-image model — more convincing.
>
> We sincerely appreciate your suggestion, which helped us improve the performance evaluation of LINA. We hope this response addresses your concerns. If there are any additional models you would like us to compare with, please feel free to let us know.
>
>
>
> **Table 1: Comparison of GenEval results.**  $^\ddagger$ denotes using rewriter, a prompt engineering method used in NOVA.
> Our \model, equipped with pure linear attention, rivals advanced T2I frameworks.
>
> | **Model**                                                    | **Params.** | **GenEval** |
> | ------------------------------------------------------------ | ----------- | ----------- |
> | NOVA (1024×1024)                                             | 1.4B        | 0.71        |
> | NOVA (512×512)$^\ddagger$                                    | 0.6B        | 0.75        |
> | Fluid                                                        | 1.1B        | 0.67        |
> | Fluid                                                        | 10.5B       | 0.69        |
> | LINA-H (1024×1024)$^\ddagger$ (1024px Training Iteration: 700k) | 1.5B        | 0.67        |
> | LINA-H (1024×1024)$^\ddagger$ (1024px Training Iteration: 50k) | 1.5B        | 0.72        |
> | LINA-H (1024×1024)$^\ddagger$ (1024px Training Iteration: 60k) | 1.5B        | 0.72        |
> | LINA-H (512×512)                                             | 1.4B        | 0.67        |
> | LINA-H (512×512)$^\ddagger$                                  | 1.4B        | 0.74        |

---

> ### Author Response · Authors · 2025-11-23
> **Response to Reviewer 7uD7 [6]**
>
> #### **Question 3: Table 3, comparing LINA’s latency to FlashAttention, reveals little actual speedup—22s vs. 20s at 1024px, with FlashAttention on the full-attention baseline—and the observed latency parity is not explained in terms of FLOPs or memory.**
>
> #### **Answer 3:**
>
> Thank you very much for this constructive reminder. As a paper that focuses on lightweight architectures, we fully agree that, beyond latency, metrics such as FLOPs and GPU memory usage are indeed important for evaluating efficiency.
> **We have completed the FLOPs and GPU memory measurements and provide the results below.**
>
> We have also updated the FLOPs-related content in **Section 5.2** accordingly. If you have any further questions or concerns, please do not hesitate to let us know. We sincerely appreciate your suggestions, as they have helped us improve our work greatly. Thank you again!
>
>
>
> **1) FLOPs Comparison: Empirical Evaluation**
>
> We compare the FLOPs of linear attention and full attention. We report the FLOPs of **a single** attention module under the following configuration: batch size of 1, sequence length of 5120, hidden dimension of 1536, and 16 attention heads. This setup matches how LINA-H operates at a resolution of 1024px.
>
> The linear attention module we evaluate adopts *division-based normalization* in the LINA *Decoder* and integrates both the *DWC module* and the *KV gate* proposed in this work. FLOPs are measured using the fvcore [1] library.
>
> The results are presented in **Table 1** below. A full-attention module requires approximately \~129 GFLOPs, whereas the linear-attention module requires only \~50 GFLOPs. This corresponds to a reduction of about **\~61%** in computation, highlighting the efficiency benefits of our LINA. Importantly, despite this substantial reduction in FLOPs, the text-to-image performance of LINA remains competitive with the full-attention-based NOVA.
>
> For clarity, we note that the *DWC module* and the *KV gate* are used only in the LINA *Decoder*, and are not applied in the *Encoder* or the *Connector*.
>
> [1] https://github.com/facebookresearch/fvcore
>
>
>
> **Table 1: FLOPs comparison results. Compared with the full attention, a single linear attention module reduces FLOPs by about ~61%, showing computation efficiency.**
>
> | Module                                     | GFLOPs |
> | ------------------------------------------ | ------ |
> | Linear Attention (w/ DWC module & KV gate) | 49.99  |
> | Full Attention                             | 128.85 |
>
>
>
> ##### **2) GPU Memory Comparison: Empirical Evaluation**
>
> We compared the GPU usage when generating a 1024px image using NOVA with **Flash Attention** and our LINA model with **linear attention**. The results are shown in **Table 2**, where we report both allocated and reserved memory. As the table shows, without applying any extra optimization techniques, our LINA model achieves GPU memory usage that is comparable to the Flash Attention version of full attention.
>
> We fully acknowledge that, in terms of GPU memory, LINA does not show a clear advantage over Flash Attention. However, we would like to gently share our perspective:
>
> Flash Attention is a **highly optimized** and mature implementation designed specifically for **full attention**, while our linear attention is implemented in plain PyTorch *without any additional acceleration*. Given this gap in engineering maturity, it is not surprising that their memory usage appears similar. In fact, this result highlights an important point: **linear attention already matches Flash Attention *before* any dedicated optimization**, showing its inherent algorithmic efficiency. Demonstrating this **algorithm-level potential** is one of the *key contributions* of our work.
>
> We believe that developing specialized acceleration techniques for linear attention is an exciting direction for future research, and we hope our findings can motivate further progress in this area.
>
>
>
> **Table 2: GPU memory comparison results.**
>
> | Model                      | Allocated  | Reserved   |
> | -------------------------- | ---------- | ---------- |
> | LINA (w/ Linear Attention) | 5406.55 MB | 8072.00 MB |
> | NOVA (w/ Flash Attention)  | 5401.96 MB | 7552.00 MB |

---

> ### Author Response · Authors · 2025-11-23
> **Response to Reviewer 7uD7 [7]**
>
> ##### **3) Computation complexity: Theoretical Analysis**
>
> In addition to the empirical results above, we also provide the following theoretical analysis to further clarify the efficiency advantages of linear attention. We hope this explanation is helpful and appreciate your careful consideration.
>
> We follow the notation in Section 2.2 in our main paper: assume the model uses $H$ attention heads to process $N=N_{\text{q}}+N_{\text{img}}$ tokens, where each token has a hidden dimension $D$, and the per-head dimension is $d$, satisfying $D = h d$.
>
>
>
> **Linear Attention**
>
> The computation of the per-token output $O^{(\texttt{d})}_i$ for linear attention with division-based normalization can be expressed as:
>
>
> $$
> O^{(\texttt{d})}_ i
> = \frac{\phi(Q_i)\left(\sum_ {j=1}^{N} \phi(K_j)^\top V_j \right)}
>        {\phi(Q_i)\left(\sum_ {m=1}^{N} \phi(K_m)^\top \right)}.
> $$
>
>
> Theoretically, for the numerator, we have:
> $$
> \underbrace{\phi(Q_i)}_ {\mathbb{R}^{1\times d}}
> \left(
> \sum_{j=1}^{N}
>     {
>     \underbrace{\phi(K_j)^\top}_ {\mathbb{R}^{d\times 1}}
>     \underbrace{V_j}_ {\mathbb{R}^{1\times d}}
>     }
> \right)
> \;\longrightarrow\;
> \mathbb{R}^{1\times d}.
> $$
>
>
> where, first, a $d \times 1$ matrix is multiplied by a $1 \times d$ matrix, which costs $\mathcal{O}(d^{2})$.
> Then, a $1 \times d$ matrix is multiplied by an a $d \times d$ matrix, which also costs $\mathcal{O}(d^{2})$.
>
> For the denominator, we have:
> $$
> \underbrace{\phi(Q_i)}_ {\mathbb{R}^{1\times d}}
> \left(
> \sum_ {m=1}^{N}
>     {
>     \underbrace{\phi(K_m)^\top}_ {\mathbb{R}^{d\times 1}}
>     }
> \right)
> \longrightarrow
> \mathbb{R}^{1\times 1}.
> $$
>
>
>
>
> where, a $1 \times d$ matrix is multiplied by a $d \times 1$ matrix, which costs $\mathcal{O}(d)$, which is negligible in the big-$\mathcal{O}$ sense.
>
> Computing the final output $O^{(\texttt{d})}_i$ also costs $\mathcal{O}(d)$, which is negligible in the big-$\mathcal{O}$ sense.
>
> Thus, the per-token complexity is at most $\mathcal{C}_{attn, token}=2d^{2}=\mathcal{O}(d^{2})$.
>
> For a single attention head processing $N$ tokens, the total cost is at most:
>
>
> $$
> \text{Linear attention, per-head:}\qquad
> \mathcal{C}_ {la, head}
> = N \mathcal{C}_ {la, token}
> = \mathcal{O}(N d^{2}).
> $$
>
>
>
>
> For the whole linear attention with $h$ heads (and hidden dimension $D = h d$), the total cost is:
> $$
> \text{Linear attention:}\qquad
> \mathcal{C}_ {la}
> = h \mathcal{C}_ {la, head}
> = \mathcal{O}(N D d).
> $$
>
>
>
>
> ##### **Comparison to Full Attention**
>
> The computational complexity of standard softmax attention is $\mathcal{C}_{fa}=O(N^2D)$, and the ratio between the two complexities is:
>
> $$
> \frac{\mathcal{C}_ {la}}{\mathcal{C}_ {fa}}
> = \frac{\mathcal{O}(N D d)}{\mathcal{O}(N^2D)}=\frac{d}{N}.
> $$
>
>
> Note that when LINA operates at 1024px, we have $N = 5120$, $D = 1536$, and $h = 16$, so $d = D/h = 96$. This gives a complexity ratio of **$d/N = 96/5120 \approx 0.019$**. Therefore, linear attention achieves a substantial reduction in computational cost, which is consistent with our measured FLOPs.

---

### Official Review · Reviewer_UJ9T · 2025-11-04

**Soundness:** 3
**Presentation:** 3
**Contribution:** 2
**Rating:** 4
**Confidence:** 3

**Summary:**

This paper tackles the problem of designing efficient linear attention mechanisms for autoregressive image generative models.
The paper introduces LINA, a text-to-image generator based solely on linear attention, capable of producing high-quality 1024×1024 images efficiently.
Through systematic experiments, they analyze key design choices—normalization strategies and depthwise convolution for locality enhancement. In addition, the KV gate is proposed for flexible memory management.
Extensive ablation and visualization studies demonstrate how these components improve scalability and sample quality.

**Strengths:**

1. This paper tackles an important problem of efficiency in autoregressive image generation, providing a viable solution based on linear attention.

2. The paper is well-structured and easy to follow.

3. Comprehensive experiments across diverse model scales and benchmarks are well-conducted and appreciated.

**Weaknesses:**

1. While the paper is strongly motivated by efficiency, the experiments mainly focus on generation performance. For example, efficiency comparisons (Table 3) are limited to latency and parameter count. Given that efficiency is the core motivation, a more in-depth analysis covering GPU memory usage, FLOPs, and runtime behavior is needed. Furthermore, the efficiency contribution of each proposed component (e.g., KV gate, depthwise convolution) should also be clarified.

2. It is unclear whether LINA truly achieves competitive performance as claimed (line 39). In Table 4, LINA-H performs noticeably worse than the baseline NOVA, especially in 1024×1024 generation, raising doubts about the claimed competitiveness.

3. Although the empirical findings are interesting, the analyses on normalization types and locality augmentation for linear attention appear generic rather than specific to autoregressive image generation. The identified trends seem consistent with existing results in other domains, thus limiting the novelty of the empirical insights.

4. If LINA’s efficiency gains are comparable to those achieved by popular acceleration techniques such as FlashAttention, practitioners may not find it worthwhile to adopt a new architecture. It would be valuable to demonstrate the synergy between LINA and such existing acceleration methods.

**Questions:**

None

---

> ### Author Response · Authors · 2025-11-23
> **Response to Reviewer UJ9T [1]**
>
> We sincerely thank you for taking the time to review our work. The questions you raised, such as how to evaluate efficiency and how to analyze the computational complexity of each component, have been very helpful for improving our paper. When we first read your comments, we also felt that these are points we should have addressed earlier. We apologize for this oversight, and we truly appreciate your feedback, which gave us the chance to add these important analyses in the updated version of the paper.
>
> Below, we list the updates we have made based on your comments:
>
> - Appendix F (in the updated PDF): Complexity Analysis of DWC Module and KV Gate (For Q1.2)
> - Section 5.2 (in the updated PDF): FLOPs Comparison (For Q1.1)
> - Appendix G (in the updated PDF): How Our Findings Relate to Prior Work on Linear Attention (For Q3)
> - Table 4 (in the updated PDF): Comparison of GenEval results. (For Q2)
> - Table 8 (in the updated PDF): Detailed configurations for LINA-H. (For Q1.2)
> - Figure 6 (in the updated PDF): FLOPs comparison results. (For Q1.1)
> - Table 10 (in the updated PDF): Training setting of our text-to-image generation LINA. (For Q2)
>
> In the updated version of the paper, we have removed the special color formatting used in the previous draft. The **blue** highlights indicate content that was updated during the discussion stage. Next, we respond to your questions one by one.

---

> ### Author Response · Authors · 2025-11-23
> **Response to Reviewer UJ9T [2]**
>
> #### **Question 1.1: Efficiency comparisons (Table 3) are limited to latency and parameter count. A more in-depth analysis covering GPU memory usage, FLOPs, and runtime behavior is needed.**
>
> #### **Answer 1.1:**
>
> Thank you very much for this constructive reminder. As a paper that focuses on lightweight architectures, we fully agree that, beyond latency, metrics such as FLOPs and GPU memory usage are indeed important for evaluating efficiency.
> **We have completed the FLOPs and GPU memory measurements and provide the results below.**
>
> We have also updated the FLOPs-related content in **Section 5.2** accordingly. If you have any further questions or concerns, please do not hesitate to let us know. We sincerely appreciate your suggestions, as they have helped us improve our work greatly. Thank you again!
>
>
>
> ##### **FLOPs Comparison**
>
> We compare the FLOPs of linear attention and full attention. We report the FLOPs of **a single** attention module under the following configuration: batch size of 1, sequence length of 5120, hidden dimension of 1536, and 16 attention heads. This setup matches how LINA-H operates at a resolution of 1024px.
>
> The linear attention module we evaluate adopts *division-based normalization* in the LINA *Decoder* and integrates both the *DWC module* and the *KV gate* proposed in this work. FLOPs are measured using the fvcore [1] library.
>
> The results are presented in **Table 1** below. A full-attention module requires approximately \~129 GFLOPs, whereas the linear-attention module requires only \~50 GFLOPs. This corresponds to a reduction of about **\~61%** in computation, highlighting the efficiency benefits of our LINA. Importantly, despite this substantial reduction in FLOPs, the text-to-image performance of LINA remains competitive with the full-attention-based NOVA.
>
> For clarity, we note that the *DWC module* and the *KV gate* are used only in the LINA *Decoder*, and are not applied in the *Encoder* or the *Connector*.
>
> [1] https://github.com/facebookresearch/fvcore
>
>
>
> **Table 1: FLOPs comparison results. Compared with the full attention, a single linear attention module reduces FLOPs by about ~61%, showing computation efficiency.**
>
> | Module                                     | GFLOPs |
> | ------------------------------------------ | ------ |
> | Linear Attention (w/ DWC module & KV gate) | 49.99  |
> | Full Attention                             | 128.85 |
>
>
>
> ##### **GPU Memory Comparison**
>
> We compared the GPU usage when generating a 1024px image using NOVA with **Flash Attention** and our LINA model with **linear attention**. The results are shown in **Table 2**, where we report both allocated and reserved memory. As the table shows, without applying any extra optimization techniques, our LINA model achieves GPU memory usage that is comparable to the Flash Attention version of full attention.
>
> We fully acknowledge that, in terms of GPU memory, LINA does not show a clear advantage over Flash Attention. However, we would like to gently share our perspective:
>
> Flash Attention is a **highly optimized** and mature implementation designed specifically for **full attention**, while our linear attention is implemented in plain PyTorch *without any additional acceleration*. Given this gap in engineering maturity, it is not surprising that their memory usage appears similar. In fact, this result highlights an important point: **linear attention already matches Flash Attention *before* any dedicated optimization**, showing its inherent algorithmic efficiency. Demonstrating this **algorithm-level potential** is one of the *key contributions* of our work.
>
> We believe that developing specialized acceleration techniques for linear attention is an exciting direction for future research, and we hope our findings can motivate further progress in this area.
>
>
>
>
>
>
>
> **Table 2: GPU memory comparison results.**
>
> | Model                      | Allocated  | Reserved   |
> | -------------------------- | ---------- | ---------- |
> | LINA (w/ Linear Attention) | 5406.55 MB | 8072.00 MB |
> | NOVA (w/ Flash Attention)  | 5401.96 MB | 7552.00 MB |

---

> ### Author Response · Authors · 2025-11-23
> **Response to Reviewer UJ9T [3]**
>
> #### **Question 1.2: The efficiency contribution of each proposed component (e.g., KV gate, depthwise convolution) should also be clarified.**
>
> #### **Answer 1.2:**
>
> We sincerely appreciate your valuable suggestion!
> Since DWC and the KV gate are key components introduced by our method, providing a clear computational complexity analysis is indeed necessary.
> We include a detailed analysis of both modules here, and we have also updated **Appendix F** of the paper accordingly.
>
>
>
> ##### **Complexity Analysis**
>
> We follow the notation in Section 2.2 in our main paper: assume the model uses $H$ attention heads to process $N=N_{\text{q}}+N_{\text{img}}$ tokens, where each token has a hidden dimension $D$, and the per-head dimension is $d$, satisfying $D = h d$.
>
>
>
> **Linear Attention**
>
> The computation of the per-token output $O^{(\texttt{d})}_i$ for linear attention with division-based normalization can be expressed as:
>
>
> $$
> O^{(\texttt{d})}_ i
> = \frac{\phi(Q_i)\left(\sum_ {j=1}^{N} \phi(K_j)^\top V_j \right)}
>        {\phi(Q_i)\left(\sum_ {m=1}^{N} \phi(K_m)^\top \right)}.
> $$
>
>
> Theoretically, for the numerator, we have:
> $$
> \underbrace{\phi(Q_i)}_ {\mathbb{R}^{1\times d}}
> \left(
> \sum_{j=1}^{N}
>     {
>     \underbrace{\phi(K_j)^\top}_ {\mathbb{R}^{d\times 1}}
>     \underbrace{V_j}_ {\mathbb{R}^{1\times d}}
>     }
> \right)
> \longrightarrow
> \mathbb{R}^{1\times d}.
> $$
>
> where, first, a $d \times 1$ matrix is multiplied by a $1 \times d$ matrix, which costs $\mathcal{O}(d^{2})$.
> Then, a $1 \times d$ matrix is multiplied by an a $d \times d$ matrix, which also costs $\mathcal{O}(d^{2})$.
>
> For the denominator, we have:
> $$
> \underbrace{\phi(Q_i)}_ {\mathbb{R}^{1\times d}}
> \left(
> \sum_ {m=1}^{N}
>     {
>     \underbrace{\phi(K_m)^\top}_ {\mathbb{R}^{d\times 1}}
>     }
> \right)
> \longrightarrow
> \mathbb{R}^{1\times 1}.
> $$
>
>
>
>
> where, a $1 \times d$ matrix is multiplied by a $d \times 1$ matrix, which costs $\mathcal{O}(d)$, which is negligible in the big-$\mathcal{O}$ sense.
>
> Computing the final output $O^{(\texttt{d})}_i$ also costs $\mathcal{O}(d)$, which is negligible in the big-$\mathcal{O}$ sense.
>
> Thus, the per-token complexity is at most $\mathcal{C}_{attn, token}=2d^{2}=\mathcal{O}(d^{2})$.
>
> For a single attention head processing $N$ tokens, the total cost is at most:
>
>
> $$
> \text{Linear attention, per-head:}\qquad
> \mathcal{C}_{attn, head}
> = N \mathcal{C}_{attn, token}
> = \mathcal{O}(N d^{2}).
> $$
>
>
>
>
> For the whole linear attention with $h$ heads (and hidden dimension $D = h d$), the total cost is:
> $$
> \text{Linear attention:}\qquad
> \mathcal{C}_{attn}
> = h \mathcal{C}_{attn, head}
> = \mathcal{O}(N D d).
> $$
>
>
>
>
> ##### **Depthwise Convolution**
>
> The DWC module applies a $k \times k$ depthwise convolution to $N_{\text{img}}$ image tokens only, with a cost of:
> $$
> \text{DWC module:}\qquad
> \mathcal{C}_{dwc}
> = \mathcal{O}(N_{\text{img}} D k^2).
> $$
>
>
> The ratio between the computational cost of the **DWC module** and that of **linear attention** is:
> $$
> \frac{\mathcal{O}(N_{\text{img}} D k^2)}{\mathcal{O}(N D d)}
> \leq
> \frac{k^2}{d}.
> $$
>
>
> Note that the DWC module processes only $N_{\text{img}}$ image tokens, while the linear attention processes both $N_{\text{q}}$ query tokens and $N_{\text{img}}$ image tokens ($N=N_{\text{q}}+N_{\text{img}}$).
>
> For our text-to-image model LINA-H, we have $D = 1536$ and $h = 16$, so the per-head dimension is $d = 96$. Therefore, we obtain $k^{2}/d = 0.26$ (DWC kernel size $k=5$).
>
> Note that the actual cost ratio is **smaller** than $0.26$.
> The reasons are as follows.
>
> **Table 3** below provides detailed configurations of LINA.
> As shown, the DWC module is used only in the *Decoder* blocks, and these Decoder blocks constitute *only $\frac{1}{3}$ of the total number of blocks*. Therefore, the DWC module introduces only non-dominant additional computational cost.

---

> ### Author Response · Authors · 2025-11-23
> **Response to Reviewer UJ9T [4]**
>
> ##### **KV Gate**
>
> The KV gate $g^{(k)},g^{(v)}\in\mathbb{R}^{N_{\text{img}}}$ uses learnable parameters to scale the keys and values on a per-token basis.
> $$
> \tilde{K}_ j=\underbrace{g_j^{(k)}}_ {\mathbb{R}^{1\times 1}}
> \underbrace{\phi(K_j)}_ {\mathbb{R}^{1\times d}},\qquad
> \tilde{V}_ j=\underbrace{g_j^{(v)}}_ {\mathbb{R}^{1\times 1}}
> \underbrace{V_j}_ {\mathbb{R}^{1\times d}}.
> $$
> For a single head processing $N_{\text{img}}$ tokens, the cost is at most:
> $$
> \text{KV gate, per-head:}\qquad
> \mathcal{C}_ {kvgate, head}
> = 2 N_ {\text{img}} d
> = \mathcal{O}(N_ {\text{img}} d).
> $$
>
>
> For the whole linear attention with $h$ heads (and hidden dimension $D = h d$), the total cost is:
>
>
> $$
> \text{KV gate:}\qquad
> \mathcal{C}_ {kvgate}
> = h \mathcal{C}_ {kvgate, head}
> = \mathcal{O}(N_ {\text{img}} D).
> $$
>
>
> The ratio between the computational cost of the *KV gate* and that of *linear attention* is:
> $$
> \frac{\mathcal{O}(N_{\text{img}} D)}{\mathcal{O}(N D d)}
> \leq
> \frac{1}{d}.
> $$
>
>
> Note that the KV gate processes only $N_{\text{img}}$ image tokens, while the linear attention processes both $N_{\text{q}}$ query tokens and $N_{\text{img}}$ image tokens ($N=N_{\text{q}}+N_{\text{img}}$).
>
> Since the per-head dimension is $d = 96$, the additional computational cost introduced by the KV gate is negligible.
>
>
>
> **Table 3: Detailed configurations for LINA-H.  The DWC module and the KV gate are used only in the LINA *Decoder*, introducing only a small amount of additional parameters and computation.**
>
> | **Structure** | Connector | Encoder | Decoder |
> | ------------- | --------- | ------- | ------- |
> | Block Number  | 16        | 16      | 16      |
> | Block Number  | 16        | 16      | 16      |
> | Block Number  | 16        | 16      | 16      |
> | Channels      | 1536      | 1536    | 1536    |
> | DWC Module    | ✗         | ✗       | ✓       |
> | KV Gate       | ✗         | ✗       | ✓       |

---

> ### Author Response · Authors · 2025-11-23
> **Response to Reviewer UJ9T [5]**
>
> #### **Question 2: LINA-H performs worse than the baseline NOVA, especially in 1024×1024 generation, raising doubts about the claimed competitiveness**.
>
>
>
> #### **Answer 2:**
>
> Thank you very much for pointing out this issue and helping us reflect on the training of high-resolution (e.g., 1024px) LINA models. After submitting the draft version, we identified the core reason behind the weaker performance of our 1024px LINA-H model: **overfitting**. In the draft version, we unsuitably trained the 512$\rightarrow$1024px stage for **700k** iterations, which caused **severe overfitting**, something we had not realized in the draft version.
>
> During the rebuttal stage, we updated the LINA-H model. In this new version, we found that training the 512$\rightarrow$1024px stage for **only 50k** or **60k iterations** already leads to a large performance improvement (**GenEval 0.72 in the new version vs. 0.67 in the draft**, for 1024px), as reported in **Table 4** of the updated manuscript.
> The settings such as learning rate and architecture remained unchanged. Moreover, the 1.4B LINA model reaches a 0.74 GenEval score at 512px, outperforming the 10.5B Fluid model and matching the 512px NOVA model with full attention (**0.74 vs. 0.75**, for 512px). These results support our conclusions about linear attention and show that LINA—a model relying entirely on linear attention—can remain competitive in text-to-image generation. We have updated the paper with all new results.
>
> While simply fixing the overfitting issue already brings a large performance gain, we believe the potential of LINA—and linear attention more broadly—extends further. With more high-quality data and stronger post-training methods (e.g., GRPO), LINA’s image quality and text alignment could likely be improved even more, though these directions are beyond the scope of this work. As a study exploring linear-attention architectures, we sincerely hope that the updated results provide stronger evidence.
>
> We would like to sincerely thank you again for your criticism. It helped us identify the serious overfitting issue and ultimately led to meaningful improvements in LINA’s performance. We also wish to kindly note that in the updated version, the high-resolution (e.g., 1024px) results have improved substantially. We hope that our solution and the updated findings can address your concerns.
>
>
>
> **Table 4: Comparison of GenEval results.**  $^\ddagger$ denotes using rewriter, a prompt engineering method used in NOVA.
> Our \model, equipped with pure linear attention, rivals advanced T2I frameworks.
>
> | **Model**                                                    | **Params.** | **GenEval** |
> | ------------------------------------------------------------ | ----------- | ----------- |
> | NOVA (1024×1024)                                             | 1.4B        | 0.71        |
> | NOVA (512×512)$^\ddagger$                                    | 0.6B        | 0.75        |
> | Fluid                                                        | 1.1B        | 0.67        |
> | Fluid                                                        | 10.5B       | 0.69        |
> | LINA-H (1024×1024)$^\ddagger$ (1024px Training Iteration: 700k) | 1.5B        | 0.67        |
> | LINA-H (1024×1024)$^\ddagger$ (1024px Training Iteration: 50k) | 1.5B        | 0.72        |
> | LINA-H (1024×1024)$^\ddagger$ (1024px Training Iteration: 60k) | 1.5B        | 0.72        |
> | LINA-H (512×512)                                             | 1.4B        | 0.67        |
> | LINA-H (512×512)$^\ddagger$                                  | 1.4B        | 0.74        |

---

> ### Author Response · Authors · 2025-11-23
> **Response to Reviewer UJ9T [6]**
>
> #### **Question 3: Although the empirical findings are interesting, the analyses on normalization types and locality augmentation for linear attention appear generic rather than specific to autoregressive image generation. The identified trends seem consistent with existing results in other domains, thus limiting the novelty of the empirical insights.**
>
> **Answer 3:**
>
> Thank you very much for this insightful comment. We fully acknowledge that neither the normalization types used in linear attention nor the locality augmentation techniques are our original inventions. Your assessment is absolutely correct. However, to the best of our knowledge, our work is the *first* to systematically investigate these components in the context of **autoregressive image generation**. Prior studies such as *Flattened Transformer* [1] and *InLine* [2] only explored them in perception tasks such as image classification.
>
> To clarify the relationship between our findings and existing results in other domains, we summarize the key similarities and differences below.
>
> ##### Differences from Visual Perception Tasks
>
> * In autoregressive image generation, our results show that **division-based normalization** yields better *scaling behavior* for linear attention than **subtraction-based normalization**. We would like to point out that *InLine* [2], in **vision perception tasks**, adopts subtraction-based normalization. In contrast, by analyzing the scaling behavior, our work recommends using division-based normalization for **autoregressive image generation**. As a result, our design choices are not fully aligned with those used in vision perception tasks. We suspect that this discrepancy may arise from the inherent gap between generation and perception tasks. Issues such as “semantic confusion”, highlighted by *InLine*, may not be the main bottleneck in generative models. Instead, division-based normalization might be inherently more suitable for denoising-based generative processes. We admit that we do not yet have a theoretical explanation for this phenomenon. Nonetheless, we believe that the empirical results and the new data points we provide will be valuable for guiding future work.
>
> ##### Similarities to Visual Perception Tasks
>
> * Consistent with findings in perception tasks (e.g., *Flattened Transformer*), our autoregressive generation results confirm that **linear attention indeed benefits from additional locality modeling**. This can be seen in the scaling behavior in Fig. 4-(a) in the main paper.
> * Nevertheless, our work further raises two conceptual questions in Sec. 3.2.2:
>
>   1. Whether the softmax operation is the key reason behind the locality gap between full attention and linear attention; and
>   2. Whether locality modeling universally helps autoregressive image generation.
>      Although our conclusions overlap with prior work in image classification, we believe these discussions offer useful conceptual insights into the role of locality in linear attention for autoregressive generation.
>
> In summary, our results suggest that **linear attention in autoregressive image generation comes with task-specific considerations**, such as preferring division-based normalization, rather than directly inheriting conclusions from perception tasks. We hope our study can provide reliable guidelines for future research, helping to avoid large amounts of repetitive ablations.
>
> Finally, we sincerely appreciate your timely and perceptive suggestion. It reminded us that these similarities and differences should be more clearly articulated in the paper. We have now added a detailed explanation in **Appendix H**. Thank you again for your thoughtful guidance, and please do not hesitate to reach out—we would be grateful for any further feedback or criticism.
>
>
>
>
> [1] FLatten Transformer: Vision Transformer using Focused Linear Attention
>
> [2] Bridging the Divide: Reconsidering Softmax and Linear Attention

---

> ### Author Response · Authors · 2025-11-23
> **Response to Reviewer UJ9T [7]**
>
> #### **Question 4: If LINA’s efficiency gains are comparable to those achieved by popular acceleration techniques such as FlashAttention, practitioners may not find it worthwhile to adopt a new architecture. It would be valuable to demonstrate the synergy between LINA and such existing acceleration methods.**
>
> #### **Answer 4:**
>
> Thank you very much for your thoughtful comments. We fully agree with your observation. As our released code shows, the current PyTorch-native implementation of linear attention does not outperform full attention equipped with mature acceleration techniques. We openly acknowledge this limitation, as also reflected in Table 3 of the main paper.
>
> That said, we would like to gently highlight one encouraging finding. Even without any specialized acceleration, our LINA—implemented purely in vanilla PyTorch—already achieves a speed close to full attention supported by FlashAttention. We see this as evidence of the strong potential of linear attention itself. This promising result mainly comes from our algorithmic contributions: exploring how to design linear attention specifically for autoregressive image generation. Based on our study, we recommend: (1) using division-based normalization together with a lightweight convolution module to recover local modeling capacity, and (2) adding the proposed KV Gate to further strengthen linear attention. We hope these insights can serve as a foundation for future work on more advanced and mature acceleration techniques. Hardware-level optimization is undoubtedly important, but it falls somewhat outside the scope of our present study.
>
> You also asked whether practitioners will adopt our architecture. This is indeed an important question. We agree that industry teams driven by product constraints may not immediately deploy our method, although several successful LLMs already employ linear attention (e.g., Qwen3-Next [1], Kimi-Linear [2]). We believe linear attention will become a competitive choice for future image generation models as well. Moreover, we are confident that researchers in the academic community and practitioners who focus on model efficiency will find LINA valuable: our work completes the core algorithmic design, and the next natural step is to pair it with stronger acceleration techniques—something many practitioners care deeply about.
>
> Finally, regarding whether linear attention can synergize with existing acceleration methods: our answer is yes. In the LLM community, this synergy has already been demonstrated—for instance, flash-linear-attention [3] offers a Triton-based implementation enabling efficient training and inference. Likewise, linear-attention models such as Qwen3-Next are already supported by high-performance inference engines like vLLM [4]. We share the view that linear attention has great potential. Our work is the first to explore how to properly design it for the relatively under-explored domain of autoregressive image generation. We hope LINA can serve as a starting point that inspires and facilitates future research on efficiency-oriented model design.
>
>
>
> [1] https://qwen3-next.com/zh
> [2] Kimi Linear: An Expressive, Efficient Attention Architecture
> [3] https://github.com/fla-org/flash-linear-attention
> [4] https://qwen.ai/blog?id=4074cca80393150c248e508aa62983f9cb7d27cd&from=research.latest-advancements-list

---

### Author Response · Authors · 2025-11-29
**General Response to ACs, SACs, and PCs**

**Dear ACs, SACs, and PCs,**

We would like to express our deepest gratitude for your time and for the constructive insights provided by the reviewers. Our initial scores (6, 4, 4, 4) make it clear that our manuscript may not have been immediately compelling, and we sincerely appreciate the careful consideration that each reviewer has given. Among the raised concerns, we believe two recurring issues deserve your particular attention.

------

### **1. GPU Memory Usage (and FLOPs Where Relevant)**

Reviewers **UJ9T** and **7uD7** expressed concerns regarding both FLOPs and memory usage, while **vP1E** specifically raised concerns about memory. In the updated manuscript, we have directly and transparently addressed these issues.

In **Section 5.2** and the **discussion**, we provide clear comparisons of FLOPs and GPU memory. Our findings show:

- **LINA’s linear attention reduces FLOPs by 61%** compared to full attention.
- **LINA’s GPU memory footprint is comparable to NOVA**, which uses flash attention.

We have not avoided or softened these concerns; instead, we have addressed them head-on, with concrete data. We hope this enables the community to confidently adopt LINA as an efficient and reliable baseline for autoregressive image generation.

For your convenience, we summarize the updates made during the discussion phase:

- **Appendix F**: Complexity analysis of the DWC module and KV Gate
- **Section 5.2**: FLOPs comparison
- **Appendix G**: Relation to prior work on linear attention
- **Table 4**: GenEval comparison
- **Table 8**: Detailed configurations for LINA-H
- **Figure 6**: FLOPs comparison results
- **Table 10**: Training settings for our text-to-image LINA

------

### **2. Novelty of Our Contribution**

Reviewers **7uD7**, **wwwH**, and **vP1E** all raised concerns about novelty. We understand and respect these concerns deeply. Here, we want to provide a clear, honest, and direct explanation of our contribution.

Our work is the **first to systematically study how to design linear attention for autoregressive image generative models with continuous tokens**. Concretely:

- We empirically explore two major design choices in this new setting:
   (1) **Which normalization paradigm to use** for linear attention (division-based vs. subtraction-based), and
   (2) **Whether additional locality modeling is beneficial**.
- We introduce the **KV Gate**, a simple yet powerful mechanism that enables flexible memory management by directly modulating keys and values. This is especially meaningful for autoregressive models requiring **bidirectional linear attention**, and our detailed ablations support its effectiveness.
- We deliver **LINA**, a computationally efficient and strong text-to-image baseline that we hope can support and accelerate future research on lightweight generative models.

We have approached these questions with sincerity, transparency, and respect for the reviewers’ expertise. We truly hope our clarifications and results allow you to re-evaluate the novelty and significance of our contributions.

------

Thank you again for the time, care, and expertise you have devoted to our submission. We are deeply grateful, and we sincerely hope that our responses, the updated manuscript, and the contributions of LINA can help inform your judgment.

With appreciation and respect,

 **The Authors**

---

### Meta-Review · Area_Chair_3Y4p · 2026-01-18

**Summary:**

Overall, reviewers agree that the experiments are extensive. Some initial concerns were centered on efficiency and novelty. The rebuttal addressed most concerns about efficiency clarification, but the fundamental problems of novelty remain mostly unchanged.

**Reviewer Concerns:**

The major concern after rebuttal might still center on the fact that the novelty remains incremental. Multiple reviewers might still view the method as a careful empirical combination of known components.

**Reviewer Scores:**

The reviewers' score is likely to center around borderline, and is less likely to strongly champion this work.

---

### Decision · Program_Chairs · 2026-01-26

Reject